# Horizontal transmission and recombination maintain forever young bacterial symbiont genomes

**Shelbi L. Russell**[1,2]\*, **Evan Pepper-Tunick**[2,3], **Jesper Svedberg**[2,3], **Ashley Byrne**[1], **Jennie Ruelas Castillo**[1], **Christopher Vollmers**[2,3], **Roxanne A. Beinart**[4], **Russell Corbett-Detig**[2,3]\*

**1** Department of Molecular Cellular and Developmental Biology. University of California Santa Cruz, Santa Cruz, California, United States of America, **2** Department of Biomolecular Engineering. University of California Santa Cruz, Santa Cruz, California, United States of America, **3** Genomics Institute, University of California, Santa Cruz, California, United States of America, **4** Graduate School of Oceanography. University of Rhode Island, Narragansett, Rhode Island, United States of America

\* shelbilrussell@gmail.com (SLR); russcd@gmail.com (RCD)

**Data Availability Statement:** Data and genome assemblies generated in this study are available through NCBI BioProject number PRJNA562081 (BioSample numbers listed in S1 Table). Code

## Abstract

Bacterial symbionts bring a wealth of functions to the associations they participate in, but by doing so, they endanger the genes and genomes underlying these abilities. When bacterial symbionts become obligately associated with their hosts, their genomes are thought to decay towards an organelle-like fate due to decreased homologous recombination and inefficient selection. However, numerous associations exist that counter these expectations, especially in marine environments, possibly due to ongoing horizontal gene flow. Despite extensive theoretical treatment, no empirical study thus far has connected these underlying population genetic processes with long-term evolutionary outcomes. By sampling marine chemosynthetic bacterial-bivalve endosymbioses that range from primarily vertical to strictly horizontal transmission, we tested this canonical theory. We found that transmission mode strongly predicts homologous recombination rates, and that exceedingly low recombination rates are associated with moderate genome degradation in the marine symbionts with nearly strict vertical transmission. Nonetheless, even the most degraded marine endosymbiont genomes are occasionally horizontally transmitted and are much larger than their terrestrial insect symbiont counterparts. Therefore, horizontal transmission and recombination enable efficient natural selection to maintain intermediate symbiont genome sizes and substantial functional genetic variation.

## Author summary

Symbiotic associations between bacteria and eukaryotes are ubiquitous in nature and have contributed to the evolution of radically novel phenotypes and niches for the involved partners. New metabolic or physiological capacities that arise in these associations are typically encoded by the bacterial symbiont genomes. However, the association itself endangers the retention of bacterial genomic coding capacity. Endosymbiont genome evolution

written and used in our analyses is available from https://github.com/shelbirussell/ForeverYoungGenomes_Russell-et-al. Underlying numerical data for all graphs and summary statistics are available as Supporting Information.

**Funding:** This work was supported by UC Santa Cruz, Harvard University, the Alfred P. Sloan Foundation (to RCD; sloan.org), and the NIH (R35GM128932 to RCD; nih.gov). Funding for Lau Basin collections was provided by the Schmidt Ocean Institute and NSF (OCE-1819530 to RB; nsf.gov), Funding for the Veatch Canyon collection was provided via a UNOLS Early Career Training Cruise Program funded by the NSF (OCE-1641453, OCE-1638805, OCE-1214335, OCE- 1655587, and OCE-1649756; nsf.gov) and the ONR (N00014–15–1–2583; onr.navy.mil). The funders had no role in study design, data collection and analysis, decision to publish, or preparation of the manuscript.

**Competing interests:** The authors have declared that no competing interests exist.

theory predicts that when bacterial symbionts become restricted to host tissues, their populations cannot remove deleterious mutations efficiently. This ultimately results in their genomes degrading to small, function-poor states, reminiscent of organellar genomes. However, many ancient marine endosymbionts do not fit this prediction, but instead retain relatively large, gene-rich genomes, indicating that the evolutionary dynamics of this process need more thorough characterization. Here we show that on-going symbiont gene flow via horizontal transmission between bivalve hosts and recombination among divergent gammaproteobacterial symbiont lineages are sufficient to maintain large and dynamic bacterial symbiont genomes. These findings indicate that many obligately associated symbiont genomes may not be as isolated from one another as previously assumed and are not on a one way path to degradation.

## Introduction

Bacterial genomes encode an enormous diversity of functions, which enable them to create radically novel phenotypes when they associate with eukaryotic hosts. However, they are at risk of genome degradation and function loss within these associations. Among the diversity of eukaryotic hosts they inhabit, symbiont genome sizes range from nearly unreduced genomes that are similar to their free-living relatives (~3–5 Mb) to highly reduced genomes that are less than 10% of the size of their free-living ancestors (~0.2–0.6 Mb) [1,2]. The degradation process often leaves genes required for the bacterium's role in the symbiosis, and removes seemingly vital genes, such as those involved in DNA repair and replication [1]. Although genome erosion may be enabled in some cases due to "streamlining" benefits [3], it is clear that the process can become problematic and can ultimately result in symbiont replacement or supplementation [1,4] to accomplish the full repertoire of functions needed in the combined organism. Thus, to fully understand how symbioses evolve, we must first understand the pressures their genomes experience.

Symbiont genome evolution theory predicts that upon host restriction, bacterial genomes begin the steady and inexorable process of decay due to decreased population sizes and homologous recombination rates, which result in inefficacious natural selection [5]. The transmission bottleneck that occurs when symbionts are passed on to host offspring further exacerbates these dynamics by making deleterious mutations more likely to drift to fixation in the next generation [5,6]. Indeed, many endosymbiont taxa exhibit weak purifying selection at the gene level [7–10]. In the early stages of bacterial symbiont-host associations, deleterious mutations arise on each symbiont chromosome and a portion drift or hitchhike with adaptive mutations to fixation. Subsequently, pseudogenes are lost entirely via deletion [5]. Ultimately, this process is thought to result in an organelle-like genome that is a fraction of the size of its free-living ancestors and has relegated many of its core cellular functions (*e.g.*, DNA replication and repair, cell wall synthesis, etc.) to the host or lost them entirely.

While the degraded genomes of many endosymbionts are consistent with this general theory, such as symbionts of terrestrial sap and xylem-feeding insects [1], a diversity of associations present discrepancies. In particular, most known marine obligate endosymbionts' genomes are at least one megabase in size and contain diverse coding content [11–17], although more reduced representatives have recently been found [18]. Large symbiont genomes in obligate associations are usually interpreted as reflecting the earliest stages of genome degradation [19,20], which is surprising considering the antiquity of many of these associations (*e.g.*, [21,22]). Furthermore, even the vertically transmitted marine symbionts

whose phylogenies mirror those of their hosts have only partially degraded genomes [23]. While important genes have been lost in some lineages, such as the recombination and repair gene *recA* and transversion mismatch-repair gene *mutY* [24], their loss did not portend an organelle-like fate.

Symbioses in marine environments exhibit significantly more horizontal transmission between hosts than those in terrestrial environments [25], suggesting that symbiont gene flow between hosts may prevent genome decay by enabling high rates of homologous recombination, efficient natural selection, and the maintenance of highly diverse genome contents. This hypothesis has so far not been tested and it is not known whether marine endosymbionts represent the early stages of genome degradation, as implied by the canonical endosymbiont genome theory, or if on-going gene flow and recombination can stall such genome decay over evolutionary time. Furthermore, although the general process of symbiont genome degradation is well-understood in theory (*e.g.*, [1,2,26]), no empirical study has directly evaluated the role of horizontal transmission and recombination in facilitating efficient natural selection and, thereby, the suppression of degradation. Interspecific-population level comparisons are essential for testing these important and long-standing questions [26].

To determine which evolutionary forces prevent marine endosymbiont genomes from degrading despite host restriction, we leveraged both population and comparative genomics of six marine bacterial-animal symbioses from three host taxa that exhibit modes of transmission across the spectrum from strict horizontal transmission (*Bathymodiolus* mytilid mussels) [27–29], to mixed mode transmission (solemyid bivalves) [15,30–32], to nearly strict vertical transmission (vesicomyid clams) [23,33–35] (see Fig 1A and S1–S3 Tables). Each of these three groups evolved independently (see Fig 2) to obligately host either a single intracellular gammaproteobacterial symbiont 16S rRNA phylotype within their gill cells or two or more phylotypes, in the case of some mussels [36,37]. These symbionts provide chemosynthetic carbon fixation, either through sulfide or methane-oxidation, to nutritionally support the association [25]. Hosts are nearly completely dependent on symbiont metabolism [38], and the solemyids have lost the majority of their digestive tracts in response [39]. These associations appear to be obligate for the symbionts as well as the hosts because either the symbionts have never been found living independently, *e.g.*, solemyid [31] and vesicomyid symbionts [40], or they have only been found in the host and surrounding environment, *e.g.*, bathymodiolin symbionts [27,41,42]. Both the vesicomyids and solemyids transmit symbionts to their offspring through allocating tens to hundreds of symbiont cells to their broadcast spawned oocytes [31,33]. While the mechanism of horizontal, host-to-host transfer has not been identified in the *Bathymodiolus* or solemyid symbionts, signatures of rampant horizontal transmission are evident in the population genetics of both of these groups [15,27,28] and the developmental biology of *Bathymodiolus* [29].

This group of marine symbioses presents an ideal system in which to test the impact of transmission mode and homologous recombination rate on bacterial symbiont genome evolution. The wealth of information known about how the vesicomyid, solemyid, and bathymodiolin symbioses function and evolve makes this a powerful evolutionary model. Furthermore, the similarity of these associations to other invertebrate-bacterial associations in the marine environment, *e.g.*, in mode of reproduction (broadcast spawning), phylogeny (Mollusca-Gammaproteobacteria), immunology (innate), ancestral feeding type (filter or deposit feeders), symbiosis function (nutritional), makes this an ideal microcosm for understanding how symbiont population genetics impact the process of symbiont genome degradation. Here, we use this model study system and a powerful population genomic approach (described in S1 Fig and Sections 1-3 of S1 Text), to show that homologous recombination is ongoing even in the most strictly vertically transmitted associations and may enable the maintenance of large and intermediate genome sizes indefinitely.

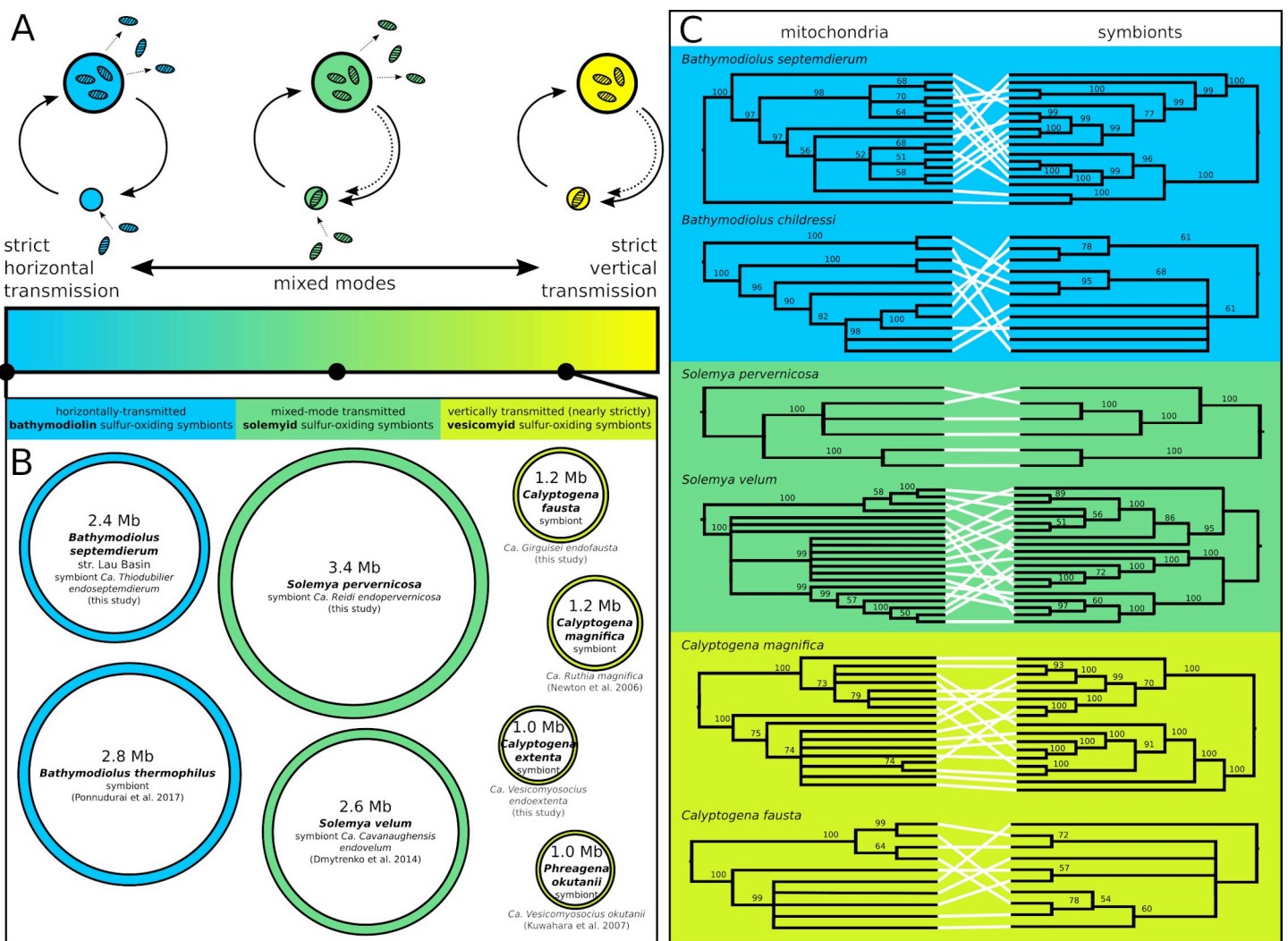

**Fig 1. Nearly-strictly vertically transmitted chemosynthetic endosymbionts exhibit genome erosion despite ongoing horizontal transmission events in their populations.** A) Transmission mode spectrum from strict horizontal transmission to strict vertical transmission, with a diversity of mixed modes, incorporating both strategies, in between. B) Genome sizes from this and previous studies [11–13,92] reveal consistent patterns of moderate genome erosion among the vesicomyid symbionts, but not in the other groups with higher rates of horizontal transmission. C) Mitochondrial and symbiont whole genome genealogies are discordant for all groups, indicating that sufficient amounts of horizontal transmission occur in vertically transmitted vesicomyid populations to erode the association between these cytoplasmic genomes. Maximum likelihood cladograms are midpoint rooted, and nodes below 50% bootstrap support are collapsed. Species are color coded by their symbiont transmission mode as in A).

## Results and discussion

Comparative analyses of host mitochondrial and endosymbiont genome genealogies show strong evidence for horizontal transmission in all six populations. The genealogical discordance shown in Fig 1C indicates that horizontal transmission has occurred in the histories of all six of these populations, however, it does not suggest how much because concordance is eroded even by exceedingly low rates of horizontal transmission [43], saturating the signal genealogies can provide. Despite this apparent similarity in transmission mode, the vesicomyid endosymbiont genomes are approximately one half the size of the solemyid or mytilid endosymbiont genomes (Fig 1B), which themselves are approximately consistent with their free-living ancestors. Nonetheless, at 1–1.2 Mb, the partially degraded vesicomyid endosymbiont genomes are still ten times larger than the smallest terrestrial endosymbionts [1]. While

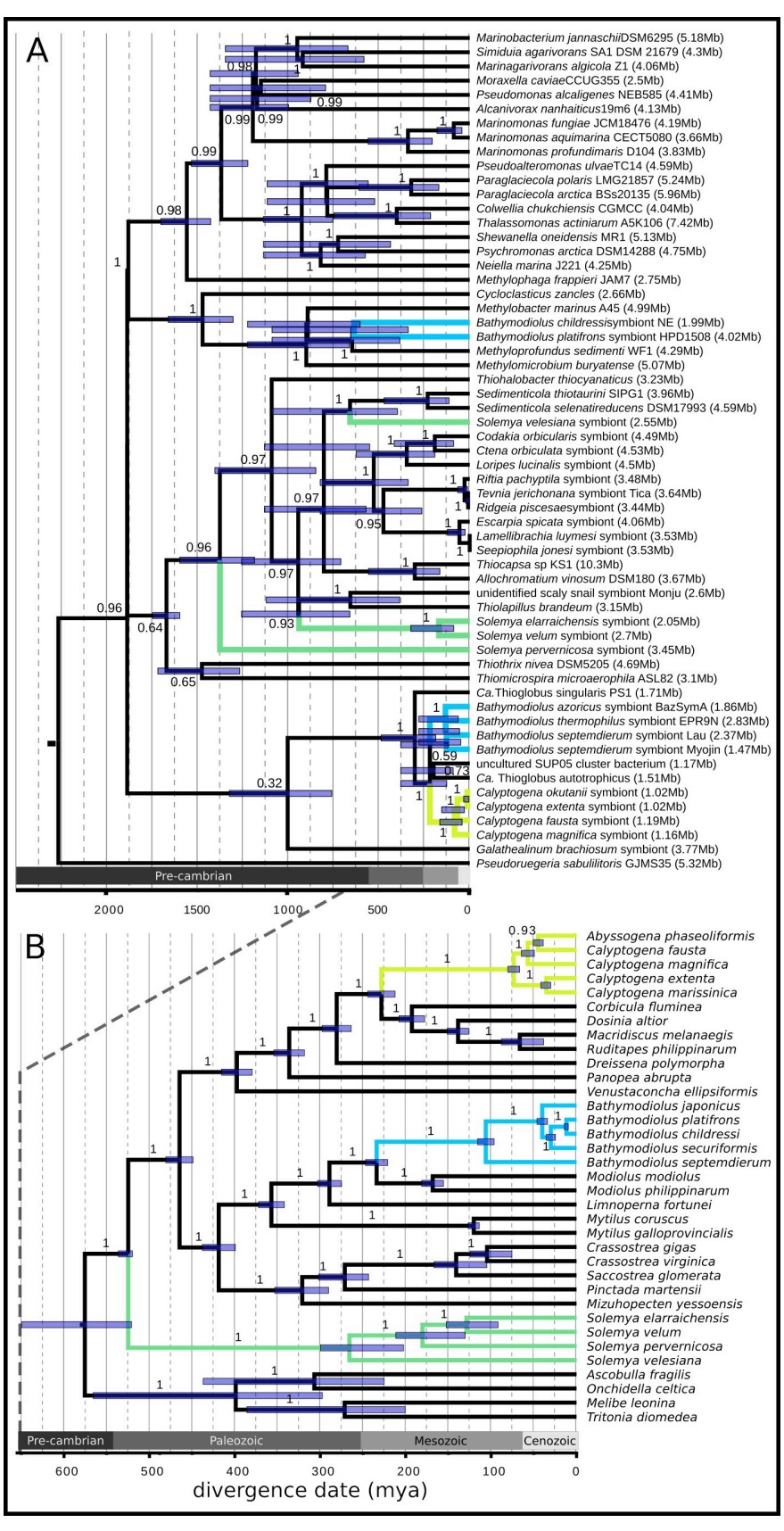

**Fig 2. Chemosynthetic bacterial symbionts and their bivalve hosts exhibit ancient divergence times.** A) Maximum likelihood phylogeny inferred from 108 orthologous protein coding genes and the 16S and 23S rRNA genes (outgroup = Alphaproteobacteria; branch labels = bootstrap support fraction) with RelTime divergence date estimates (node bars = 95% confidence intervals). Host-associated bacteria are listed as symbionts of their host species. Bacterial genome sizes are written to the right of the taxon names in the tip labels to highlight trends in genome size across clades. B) Whole mitochondrial Bayesian phylogeny for bivalves (outgroup = Gastropoda; branch labels = posterior probabilities) with divergence dates co-inferred in Beast2 (node bars = 95% highest posterior densities). In both phylogenies, members of vesicomyid, solemyid, and bathymodiolin (both thioautotrophic and methanotrophic) associations are colored yellow, green, and blue, respectively.

the fossil record and previous phylogenetic analyses indicate that the vesicomyid symbionts and clades of solemyid symbionts have been in continuous host association for long periods of time [21,22,44], and thus genome erosion has been prevented, precise divergence dates were needed to confirm this.

Divergence date estimates for hosts and symbionts indicate that the observed patterns in symbiont genome size have been maintained over many millions of years (Fig 2, S2 Fig, and S4–S6 Tables). Similar to prior work [21], we estimated that the vesicomyid bivalves evolved from their non-symbiotic ancestors around 73 million years ago (mya) (95% highest posterior density (HPD) = 62.59–76.70 mya; Fig 2B). We estimated a similar divergence date for the vesicomyids' monophyletic symbionts of around 84 mya (95% CI = 42.37–165.09 mya; Fig 2A), which is remarkable given that their loss of DNA repair genes and reduced selection efficacy has likely increased their substitution rate (and may have been accounted for by using a relaxed local molecular clock). The clade of gammaproteobacteria that contains the vesicomyid symbionts, termed the SUP05 clade, also contains the thiotrophic *Bathymodiolus* symbionts and free-living bacteria with genomes in the range of 1.17–1.71 Mb (Fig 2A), from which the vesicomyid symbionts are approximately 220 mya (95% CI = 127 - 381mya) diverged. This indicates that the vesicomyid symbiont genomes have eroded 58% at most, depending on the symbiont lineage and the ancestral state. Thus, over tens of millions of years of host association, the vesicomyid symbiont genomes have exhibited high degrees of stasis and have only degraded moderately (*e.g.*, in vesicomyid symbionts with 1 Mb vs. 1.2 Mb genomes).

The solemyids present a similar, but more complicated situation, likely owing to their antiquity, as the hosts first appeared in the fossil record more than 400 mya [22], when the ocean basins had much different connectivity [45]. While host-switching and novel (free-living) symbiont acquisition have certainly occurred in Solemyidae, and our data indicates such an event may have happened after *S. velum* and *S. pervernicosa* diverged (Fig 2A), it may be relatively rare across geological time. Whole genome mitochondrial and symbiont phylogenies indicate that the North Atlantic *Solemya* species *S. velum* and *S. elarraichensis* are sisters that diverged around 129 mya (95% HPD = 124–152 mya; Fig 2B), and their symbionts likely co-speciated with them around the same time (170 mya, 95% CI = 89–325 mya; Fig 2A). Divergence and subsequent speciation may have been due to the opening of the Atlantic, which occurred contemporaneously (180–200 mya; [46]). Thus, the vertically transmitted *S. velum* and *S. elarriachensis* symbionts have maintained genomes similar in size to their free-living relatives over hundreds of millions of years of host association.

Given the ancient ages of these associations (Fig 2) and their non-negligible rates of horizontal transmission (Fig 1C), a finer scaled exploration of symbiont genetic diversity was necessary to characterize the population-level processes influencing genome erosion. Homologous recombination is an important driver of genetic diversity in many bacterial populations [47,48], and could impact host-associated symbiont populations if diverse genotypes co-occur due to horizontal transmission. Novel genotypes are necessary because recombination among clonal strains, *e.g.*, within a host-restricted clonal population of vertically

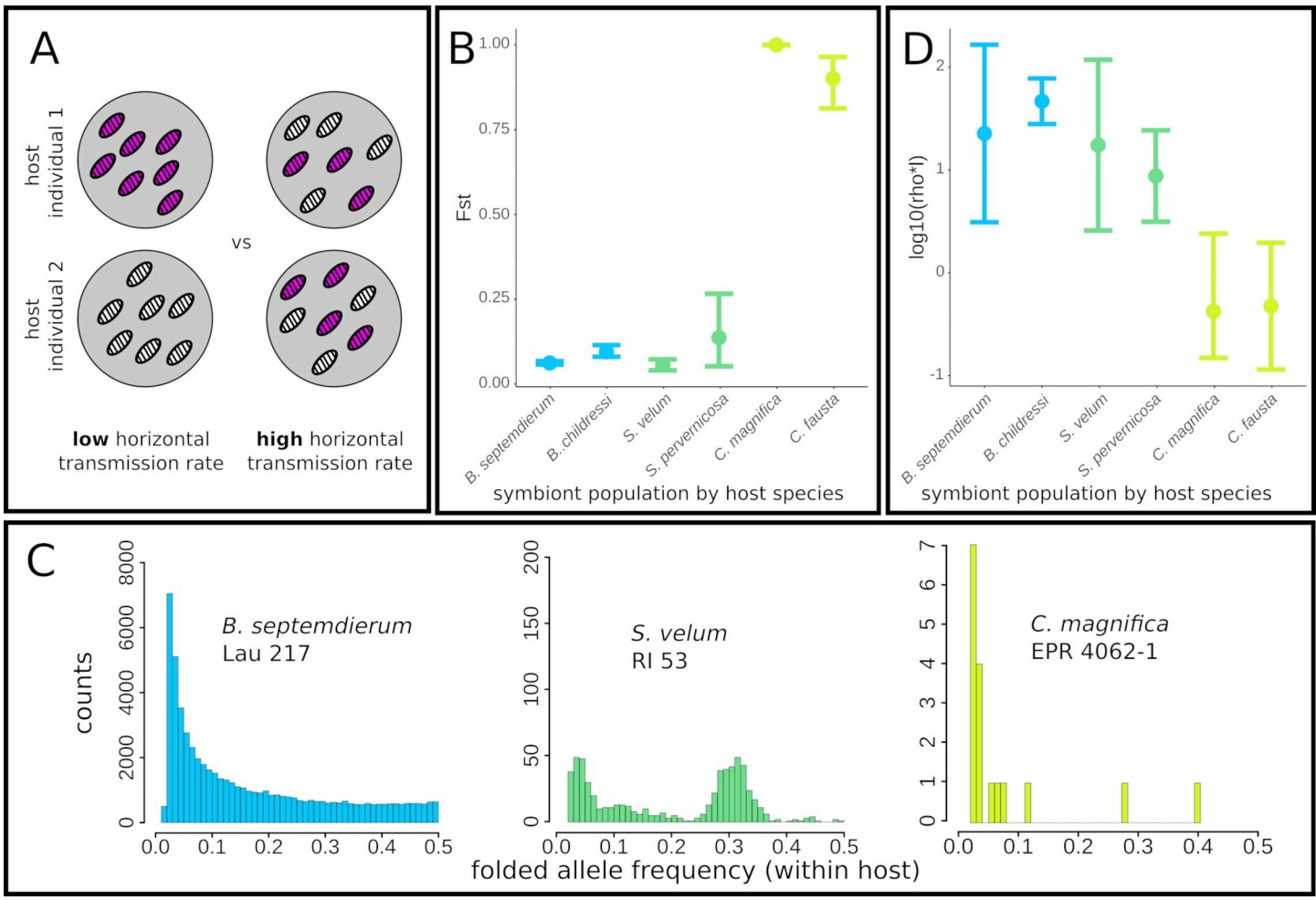

**Fig 3. Horizontal transmission and recombination introduce novel alleles into symbiont populations.** A) Model of endosymbiont genotype (pink vs. white) distributions under well-mixed high horizontal transmission rates and differentiated, low horizontal transmission rates. B) Horizontally transmitted mytilid (blue) and mixed mode transmitted solemyid (green) symbionts are well mixed among hosts, whereas the nearly strictly vertically transmitted vesicomyid symbionts (yellow) are highly differentiated among hosts. Error bars = 95% confidence intervals from non-parametric bootstrapping. C) Intrahost population folded allele frequency spectra (AFS) are shaped by access to gene flow, which is enabled by horizontal transmission and recombination. D) Recombination rates are significantly higher in the mytilid (blue) and solemyid (green) symbiont genomes compared to the vesicomyid symbiont genomes (yellow). Error bars = 95% confidence intervals.

transmitted endosymbionts, has little impact on haplotypic diversity [49]. To explore the opportunity for recombination among divergent clades of chemosynthetic symbionts, we partitioned symbiont genetic variation to between and within-host variation (*e.g.,* Fig 3A and S7 and S8 Tables). Within vesicomyid symbionts, genetic diversity is strongly subdivided by hosts, and nearly all variation distinguishes host populations (Fig 3B). Conversely, for mytilid and solemyid endosymbionts, hosts have little impact, and two endosymbionts within a host are almost as divergent as two from different hosts (Fig 3B).

The distribution of genetic diversity within a single host is even more striking than the pattern between hosts. Within host individuals, *Bathymodiolus* endosymbionts are exceptionally genetically diverse and the allele frequency spectra are qualitatively similar to expectations for an equilibrium neutrally-evolving population (Fig 3C left and S3 Fig), consistent with the high genetic diversities reported for other bathymodiolin symbionts [50]. Conversely, solemyid endosymbionts maintain intermediate and more variable within-host genetic diversity, consistent with a mixture of vertical and horizontal transmission (Fig 3C middle and S4 Fig). Finally,

vesicomyid endosymbiont populations within hosts are virtually devoid of genetic variability (Fig 3C right and S5 Fig). Therefore, despite their literal encapsulation within host cells, mytilid and solemyid symbionts have abundant opportunities to recombine and create fitter chromosomes whereas vesicomyid symbionts must only rarely encounter genetically differentiated individuals.

Although opportunities are limited for vesicomyid endosymbionts, even relatively infrequent homologous recombination events might drive patterns of genome evolution. We therefore developed a theoretical framework of symbiont evolution during mixed transmission modes. Importantly, our model demonstrates that in some conditions these populations can be approximated using a standard Kingman-coalescent and that horizontal transmission is mechanistically linked to observable recombination events between genetically diverse symbiont genomes (Section 2 of S1 Text and S6 and S7 Figs). We then performed extensive coalescent simulations and used a Random Forest-based regression framework to estimate the effective recombination rates for each population (estimated as the population-scaled recombination rate (rho) per site (l); see S9 Table and Materials and Methods). Although our model is clearly an approximation, the results are generally consistent with our prior expectations. The resulting estimated recombination rates are substantially higher in mytilid and solemyid symbionts than in vesicomyid symbionts (Fig 3D and S7 and S10 Tables), indicating that the potential for recombination within hosts is realized in these species. Given that these recombination events are occurring within symbiont populations ($\theta_{recombinant} = \theta_{genome}$), our estimate of rho*l is equivalent to r/m from previous studies of bacterial recombination rates (r/m = rho*l*$\theta_{recombinant}$/$\theta_{genome}$; see [51]). Comparing to r/m values across bacteria and archaea, which range from 0.02 to 63.6 [52], reveals that these symbiont populations have some of the largest effective recombination rates ever reported for bacteria (rho*l from the *B. septemdierum* symbionts equals 46.3, which is in the 96th percentile of previously measured rates). Despite their lack of many genes normally required for recombination [24], we found evidence for modest rates of recombination within both partially degenerate vesicomyid genomes. This capability may be enabled via "illegitimate" mechanisms, *e.g.*, RecA-independent recombination via slipped-mispairing or single strand annealing [24,53]. For all three symbiont taxa, recombination has a larger impact on genome evolution than mutation (estimated by rho*l/theta in S7 Table), in part due to the relatively low estimates of theta in the symbiont populations. Thus, these bacterial symbionts comprise what might be described as quasi-sexual, rather than clonal, populations.

A fundamental consequence of decreased homologous recombination rates for endosymbionts is that selected mutations cannot be shuffled to form higher fitness chromosomes and remain linked to neutral mutations for longer times. Ultimately, this competition among selected mutations on different haplotypes, termed clonal interference, can drive the fixation of deleterious mutations and reshape genealogies towards long terminal branches and excesses of rare alleles [54] (illustrated in Fig 4A). Similarly, even in the absence of competition among selected haplotypes, recently completed selected sweeps can reshape linked neutral genealogies if recombination is infrequent [55]. Consistent with this theory, we find abundant rare alleles in the vesicomyid symbiont genomes (Fig 4F and 4G and S7 Table; Tajima's D = -1.98 and -2.03 for *C. magnifica* and *Calyptogena fausta*, respectively), but little skew in the allele frequencies of other endosymbiont populations (Fig 4B–4E and S7 Table; D ranges from -2.03 to 1.01). Importantly, it is unlikely that differences in host species demography have driven these differences, *e.g.*, recent population expansions specifically in vesicomyid clams. In fact, we found less allele frequency skew in the mitochondrial genomes than in the vesicomyid symbionts for all species considered, and the strongest negative skew in the allele frequencies in the mitochondrial genome of *B. septemdierum* from the Lau Basin (D = -1.9, S7 Table).

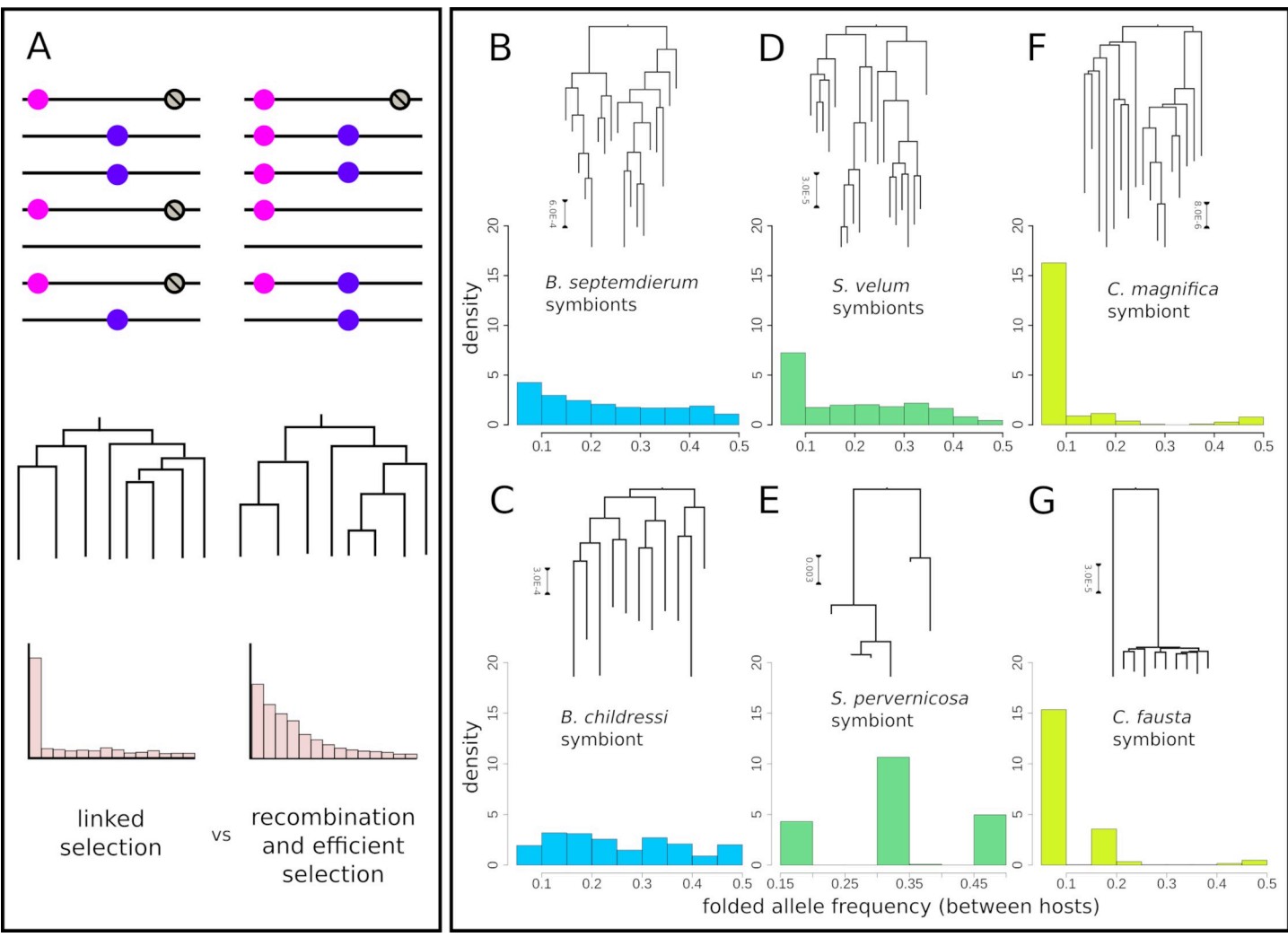

**Fig 4. Consequences of access to gene flow via horizontal transmission and recombination on the distribution of symbiont genetic diversity between hosts.** A) Diagram showing how beneficial alleles (pink) are linked to deleterious alleles (grey) in populations experiencing strong selection on linked sites versus free recombination, and how these processes are reflected in the underlying population genealogies and allele frequency spectra. B-G) Symbiont genealogies and between host allele frequency spectra (AFS) for each host/symbiont species.

Additionally, relative rates of molecular evolution between symbiont populations follow the expected trend, with dN/dS values of 0.14, 0.096, 0.083 for vesicomyid, solemyid and bathymodiolin genomes, respectively (pairwise Wilcoxon test p-values: bathymodiolin-vesicomyid p = 4.60e-14, solemyid-vesicomyid p = 1.02e-11, and bathymodiolin-solemyid p = 0.0365), consistent with recombination enabling more efficacious purifying selection for sustained periods of time.

Genome structure stasis is thought to be another hallmark of canonical endosymbiont genome evolution, and many terrestrial endosymbioses have reported static, degenerate genomes [1]. Whole genome alignments reveal that the vesicomyid symbiont genomes are highly syntenic, with few rearrangements, insertions, or deletions; whereas the other symbiont genomes are far more structurally dynamic (Fig 5 and S8 Fig). Given the role of recombination in altering bacterial genome structure ([56]; illustrated in Fig 5A), and the signals of recombination and linked selected sites in the vesicomyid symbionts (Figs 3D, 4F and 4G, respectively), recombinational processes may partially underlie genome erosion, in addition to

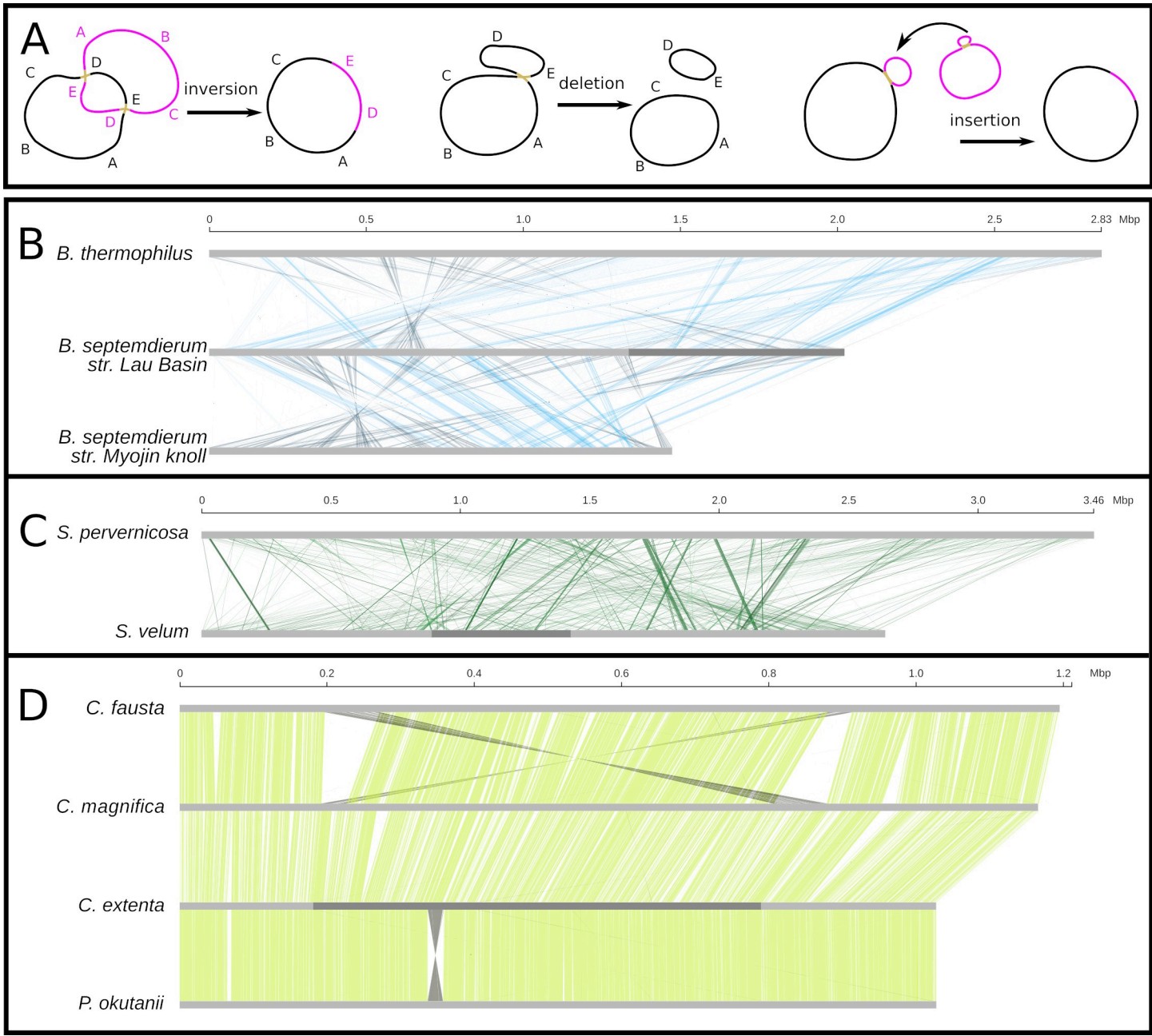

**Fig 5. Genome structure is shaped by horizontal transmission and recombination.** A) Models of recombination-based structural mutation mechanisms. B-D) Whole genome alignments for B) sulfur-oxidizing mytilid, C) solemyid, and D) vesicomyid symbiont genome assemblies with >1 Mb scaffolds.

preventing it. This could proceed in the following way: first, a rare recombination event induces a deletion that drifts to high frequency in the within-host population (other events can also induce deletions, such as strand slippage [57]). With homologous recombination events occurring so rarely in these endosymbiont genomes, natural selection would be unable to efficiently purge deleterious deletions. Although, inversions, which can be highly mutagenic by inverting the translated strand, inducing replication-transcription machinery collisions [58], are nearly absent potentially due to their high fitness costs. If symbionts with chromosomes bearing the deletion are exclusively transmitted to offspring during vertical transmission, then

the deletion would be fixed in all subsequent symbionts in that host lineage. Multiple instances of this process would incrementally reduce the size of the symbiont genome.

In contrast, the solemyid symbiont genomes exhibit increasing degrees of structural dynamics with increasing divergence time. Highly divergent solemyid symbionts, such as the *S. velum* and *S. pervernicosa* symbionts, exhibit genomes that are as structurally dynamic as strictly horizontally transmitted associations (Fig 5B vs 5C). However, over shorter time scales, such as the duration of time since the *S. velum* and *S. elarraichensis* symbionts diverged, structural changes appear to be dominated by insertions and deletions (indels; confirmed for 30 Kb segments of the *S. elarrachensis* symbiont draft genome in S7 Fig). Rapidly evolving indels have also been reported at the species level for *S. velum* [15]. Potentially underlying indel dynamics, we found that the solemyid symbionts have far more mobile genetic elements than any of the other symbiont genomes (S11 Table). High mobile element content is consistent with a combination of environmentally-exposed and host-associated periods [59], and mirrors the early stages of symbiosis [19,20], as well as the early stages of eukaryotic asexuality [60]. The mobile elements exhibit homology to different environmental bacteria, implying many independent insertion events (S12 Table).

Given the extreme ages of the solemyid and vesicomyid associations (Fig 2), our data suggest that moderate rates of recombination have allowed their symbiont genomes to maintain functional diversity characteristic of a free-living or moderately reduced genome, respectively. Evidence from the insect endosymbionts experiencing extreme genome size reduction indicates that genome erosion is possible over time frames as short as 5–20 mya (*e.g.*, [61–68]). It is plausible that vesicomyid symbionts' horizontal transmission and recombination rates are at the beginning of the range of values that permit genome decay. Our data indicates that they still undergo horizontal transmission (Fig 1C) and recombine (Fig 3D and S7 Table). Furthermore, signatures of genetic variation consistent with linked selection and sustained intermediate genome sizes indicate that selection is sufficiently efficacious to maintain some functional diversity in these populations, counter to the expectations for the original theory on endosymbiont genome evolution [5,6]. Thus, ample time has passed for these symbiont genomes to erode, but horizontal transmission and recombination have likely prevented it (as depicted in Fig 6).

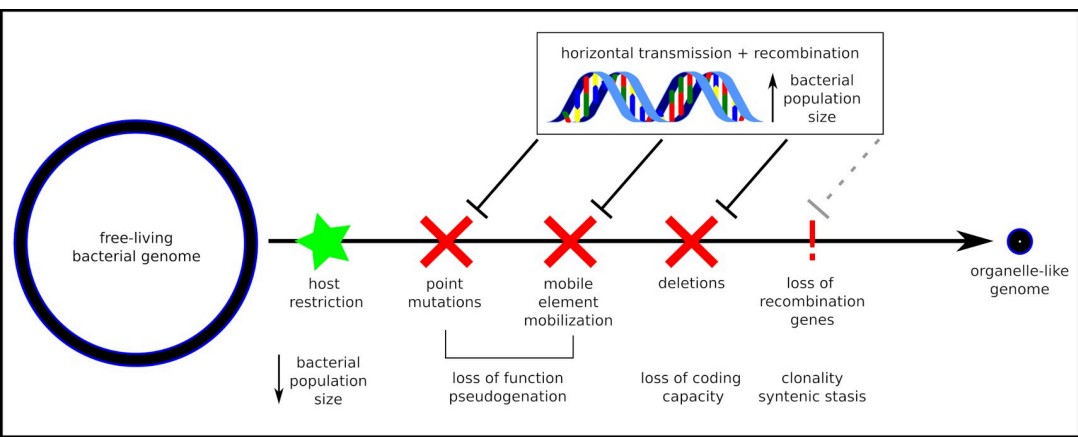

**Fig 6. A conceptual model of the prevention of endosymbiont genome degradation through horizontal transmission and recombination.** Sufficient levels of genetic diversity, which can be introduced via horizontal transmission of symbiont genotypes between hosts and recombination between genotypes in mixed infections, prevents or delays genome degradation by restoring functional versions of mutated or deleted regions. Prevention can continue until recombination capabilities (RecA-dependent and independent) are completely lost, at which point, genetic rescue is no longer possible without wholesale symbiont replacement.

## Conclusion

Here we empirically show that symbiont genome sizes and functional diversity are predicted by the rate of gene flow into and among symbiont populations via horizontal transmission and homologous recombination. Although we have only investigated three independently evolved associations, we see this system serving as a microcosm for marine associations more generally, as many other associations exhibit similar biologies (*e.g.*, intracellular, autotrophic, broadcast-spawning, etc. [36]) and all are governed by the same population genetic principles. Amazingly, we found that symbiont gene flow between hosts is ongoing in one of the most intimate marine associations, the vertically transmitted vesicomyids. These results suggest that there is a range of possible intermediate genome degradation states that can be maintained over millions of years with sufficient recombination. Therefore, symbiont genome evolution following host restriction is not a one-way, inescapable process that ends in an organelle-like state as it is commonly presented [2,5,6]. These results validate long-standing but untested theory and suggest that the diversity of symbioses found to exhibit intermediate rates of horizontal transmission and incomplete genome degradation may be undergoing similar population-level processes.

## Materials and methods

### Samples and genomic data production

**Sample collection.**   We obtained chemosynthetic bivalve samples from hydrothermal vents, cold seeps, and reducing coastal sediments from around the world (S1 Table). *Calyptogena magnifica* samples were collected from the East Pacific Rise (EPR) hydrothermal vent fields between 1998 and 2004. *Calyptogena fausta* were collected from the Juan de Fuca (JDF) Ridge hydrothermal vent system in 2004. The single *Calyptogena extenta* specimen was collected from Monterey Canyon in 1995. *Solemya pervernicosa* samples were obtained from the Santa Monica sewage outfall in 1992 (as in [15]). *Solemya velum* were collected from Point Judith, RI, as described in [15]. *B. childressi* were sampled from the Veatch Canyon cold seep off of New England. *B. septemdierum* was sampled from the ABE and Tu'i Malila hydrothermal vent sites in Lau Basin. All tissue samples were stored at -80˚C until sterile dissection or subsection of previously sterile-dissected gill tissue as described in [15].

**DNA extraction and Illumina sequencing.**   We extracted DNA from gill samples for each host individual sampled using Qiagen DNeasy kits following the manufacturer's instructions. We quantified DNA concentrations using a Qubit dsDNA kit and normalized each sample to 10 ng/ul for Illumina library preparations. We produced the majority of our Illumina sequencing libraries for each sample using a Tn5-based protocol for tagmentation followed by dual-indexing PCR using HiFi DNA polymerase (Kappa Bioscience) and custom primer sequences (IDT) designed to uniquely label both i5 and i7 indexes for each sample (Tn5 enzyme was expressed and purified in-house). Indexed samples were pooled and sequenced on single lanes of a Hiseq4000. We sequenced a total of four lanes of Hiseq4000 paired-end 150 bp sequencing across the entire study. Additionally, we obtained a subset of samples for *C. magnifica* using genomic methods from our previous work ([15], S1 Table). We also used a dataset we have previously collected for *S. velum*, specifically the population from Point Judith, Rhode Island ([15], S1 Table). The specific library preparation methods and number of read pairs obtained for each sample included in this work are listed in S1 Table.

**Nanopore sequencing.**   For each *de novo* genome assembly in this work, we selected a representative from each host population based on DNA quality as determined using an Agilent Tapestation and DNA concentration based on qubit readings. We sequenced each sample on a

single minion flow cell using the ligation-based 1D chemistry, SQK-LSK109 kit per ONT instructions with minor adjustments. The end-repair reaction was incubated for 30 minutes each at 20°C and 65°C and the ligation reaction was performed for 30 minutes instead of the recommended 10 minutes. Read counts obtained and mean read lengths are available in S1 Table. We performed basecalling using the Albacore basecaller v2.0.1 and we discarded the subset of reads whose mean quality score was less than 7. These are the set marked nominally as "failed" by the basecaller software.

***De novo* symbiont genome assembly.** Reference genomes for the *B. septemdierum*, *B. childressi*, *S. pervernicosa*, and *C. fausta* symbionts were assembled using combined Nanopore and Illumina reads. First, we assembled the Nanopore reads using the long-read assembly program wtdbg2 [69] using the "ont" presets option and setting the parameter -k to 15. Then, we performed two rounds of reference genome improvement by aligning Illumina sequencing reads from the same individual to the resulting unfiltered assembly and polishing with the Pilon software package [70]. We used BWA-MEM [71] to align Illumina reads in each subsequent polishing round.

We assembled the *C. extenta* symbiont genome from an Illumina library prepared for the single sampled individual using IDBA [72] and SPAdes [73]. While both assemblies were highly contiguous (N50 = 596007 and 604961 bp, respectively), the SPAdes assembly was able to merge two contigs that were split in the IDBA assembly, so the two contig SPAdes assembly was used for downstream analyses. Comparisons of synteny demonstrated that this join was found in other Vesicomyid endosymbiont genomes suggesting it is correct (see below).

Because read mixtures include host genomic DNA, mitochondrial DNA, and genomic DNA from other bacterial species, we then rigorously filtered the resulting contigs to extract only high confidence contigs contributed by bacteria of the study species. To identify symbiont contigs, we called ORFs with Prodigal v2.60 [74] and annotated coding sequences with BLAST [75] as described in [15] (NCBI nr, TrEMBL, and UniProt database accessed on April 7, 2019). We annotated ribosomal RNAs with RNAmmer [76] and transfer RNAs with tRNAscan [77]. Using the taxonomic information encoded in the annotation, we identified contigs that were confidently of symbiont origin. Then, using these contigs, we filtered the remaining contigs by GC content, read coverage, and coding density using custom scripts. Finally, we evaluated the quality of the assemblies with CheckM [78] and by testing for the presence of core bacterial phylogenetic markers [79] (see S2 Table).

**Host mitochondrial genome assembly.** Mitochondrial genomes were assembled from Nanopore reads, which were subsequently corrected with Illumina data, as described above, or they were assembled from Illumina reads directly. As the different samples contained different mitochondrial coverage, higher short read coverage was often better than lower long read coverage for recovering these genomes. We assembled mitochondrial genomes for *C. fausta and B. septemdierum* with IDBA [72] using Illumina data from two of the highest depth-of-coverage samples (*C. fausta* 31 and *B. septemdierum* 231, respectively). The complexity of the *B. septemdierum* data prevented IDBA from finishing within a week, so we first removed low coverage nuclear kmers (<10x) with Quake [80]. The *C. extenta* mitochondrial genome was assembled along with the symbiont genome using SPAdes [73], as described above. We were able to use the Nanopore-based assembly from *Bathymodiolus childressi* for the mitochondrial genome. Lastly, the complete mitochondrial genome for *S. pervernicosa* was available from [15], so we did not reassemble it here.

After assembly, we identified the mitochondrial scaffold by blasting the full set of scaffolds against a database containing the currently available set of 19 bivalve mitochondrial genomes. Then, we annotated the mitochondrial genome with MITOS [81]. For mitochondrial genomes lacking conserved genes, we repeated genome assembly and mitochondrial genome

identification and annotation with a different sample to verify we obtained the full sequence. See Section 1 of S1 Text for a description of the host species identification verification process.

**Short read alignment.** After producing endosymbiont and host mitochondrial genome assemblies for each host/endosymbiont species, we aligned short read Illumina data from each individual to a reference genome consisting of both of these genomes. Genomes assembled previously for the *C. magnifica* symbiont ([11], accession NC_008610.1) and mitochondrian ([82], accession NC_028724.1), and the *S. velum* symbiont ([13], accession NZ_JRAA00000000.1) and mitochondrian ([83], accession NC_017612.1) were used as references for these populations. We used the BWA mem software package [71], and we then sorted and removed duplicate reads using the samtools software package [84]. After this, we performed indel realignment for each sample separately using the "IndelRealigner" function within the Genome Analysis Toolkit (GATK) software package [85].

**Genotyping and variant filtration for each host individual.** We called consensus genotypes for each individual jointly using the GATK "UnifiedGenotyper" option and we ran the program with otherwise default parameters except we required that it output all sites rather than just all variable positions. We filtered variant sites using the vcftools software package [86] largely following the GATK best practices as we have done in our previous work [15]. Briefly, we required that each site have a minimum quality/depth ratio of 2, a maximum Fisher's strand value of 60, a minimum nominal genotype quality of 20 and a maximum number of reads with mapping quality zero at a putatively variant site of 5. For analyses of within host individual variation for fixation index (Fst) calculations, we also obtained a multiple pileup file using samtools and filtered sites that were not retained after applying these filters. See Section 1 of S1 Text for an estimate of the consensus genotyping error rate.

**Within-host diversity analysis.** We called within-host SNP and indel variants for endosymbionts and mitochondria using the method from [87]. Briefly, we created mpileup files from BWA bam alignment files for all individuals from each host species using SAMTools [84]. Then, we called variants and calculated pairwise diversity using the perl script from [87], which only considers sites within one standard deviation of the average genome coverage, filters SNPs around indels, and requires an alternate allele count in excess of the cumulative binomial probability of sequencing error at that site. As very closely related sister taxa have not been sampled for most of these bacterial genomes, ancestral/derived alleles could not be identified and we could not plot unfolded allele frequency spectra. Instead, folded allele frequency spectra were calculated for minor alleles and plotted in R.

No heteroplasmy was detected within the mitochondrial within-host populations (see S7 Table), suggesting that these bivalves do not experience double uniparental mitochondrial inheritance. This is important given our expectations regarding mitochondrial-symbiont co-divergence under strict vertical transmission.

## Genome analyses

**Population genealogy inference.** We produced multiple fasta sequence files for each population for the host mitochondrial genomes and for the concatenated symbiont genomes from the set of filtered consensus genotype calls. We then used the phylogenetic software package RAxML [88] using a GTR+G model and 1,000 bootstrap replicates to estimate the phylogenetic relationships among samples and to quantify uncertainty in our phylogenetic relationships. Using FigTree, we rooted the trees by their midpoints and created cladograms for topological comparisons.

**Analysis of polymorphic and recombinant sites.** Using the fasta files described above, we filtered sites to only retain biallelic SNPs with a minimum genotype quality of 10. Without

indels, these resequenced genomes were already aligned. Then we used the aligned SNP data to calculate Waterson's theta [89], pi [90], and the proportion of pairwise sites where all 4-gametes, *i.e.*, all pairwise combinations of alleles, are represented. We then binned the 4-gamete sites by the distances between alleles, with bins at 1e1, 1e2, 1e3, 1e4, 1e5, and 1e6 bp, for model fitting (described below).

**Whole genome structural alignment.** We generated whole genome alignment plots by first aligning bacterial genome assemblies with MUMmer 3.23 [91], using the nucmer algorithm and default parameters. In addition to the mb-scale genomes we assembled and referenced above, we obtained mb-scale genomes for *Bathymodiolus septemdierum* str. Myojin knoll ([42], accession GCA_001547755.1), *Bathymodiolus thermophilus* str. EPR9N ([92], accession GCF_003711265.1), and *Vesicomyosicus okutanii* ([12], accession NC_009465.1) from NCBI for alignment. The nucmer output was converted into the BTAB format with the MUMmer tool show-coords and was then visualized using a custom Python script. When necessary, some scaffolds in the bacterial assemblies were split into two parts in order to convert a circular genome into a linear alignment plot.

We used the whole genome aligner progressiveMauve [93] to compare genome synteny on the 10s of Kb scale between the *S. velum* and *S. elarraichensis* symbionts. First, we reordered the contigs comprising the *S. elarraichensis* symbiont draft genome [15] by the *S. velum* symbiont assembly [13] with the reorder contig function in progressiveMauve. Then, we aligned the *S. velum* symbiont genome pairwise against the *S. elarraichensis* symbiont's reordered contigs and the *S. pervernicosa* symbiont genome we assembled. We plotted the alignment backbone files in the R package genoPlotR [94].

**Mobile element analysis.** We identified mobile elements in the endosymbiont genome sequences by BLAST. First, we generated BLAST database files with the makeblastdb command from the ACLAME [95] and ICEberg [96] nucleotide and amino acid databases of transposable, viral, and conjugative elements. Next, we used blastp and blastn to compare endosymbiont amino acid sequences and full genome sequences, respectively, to these databases (cutoff values: minimum alignment length of 50 nucleotides or 50% amino acid query coverage, 90% identity, and e-value 1e-6). Overlapping hits were consolidated into a single mobile element-containing region (S12 Table).

**Ortholog identification.** We identified putative orthologous sequences among sets of bacterial genomes by a reciprocal best BLAST approach. To do this, we first performed pairwise blasts between each pair of genomes' coding sequences with blastn (-best_hit_overhang 0.1 -best_hit_score_edge 0.1 -evalue 1e-6), alternating each sequence as the query/subject. We parsed these results to only retain the best hits with >50% identity and >100 bp alignment lengths. Then, using a custom perl script, we compared hits between all pairs to identify genes with identical reciprocal best hits among all taxa each homologous gene was detected in. We used the resulting matrix of these reciprocal best hits to extract the coding sequences for each ortholog for each species from the genome fasta files for downstream analysis.

**dN/dS analysis.** To evaluate the impact of homologous recombination on patterns of natural selection at the molecular level over long periods of time we computed the average fixation rate among endosymbiont lineages at nonsynonymous and synonymous sites (dN/dS). This ratio of values is an approximate measure of the strength of purifying selection under the assumption that most nonsynonymous substitutions are deleterious. For all genes where a single ortholog was found to be shared among all symbiont lineages we began by producing codon aware alignments using MASCE [97]. Then, we compared each orthologous alignment for pairs of symbiont lineages within each group (solemyid, vesicomyid, and bathymodiolin) to estimate dN and dS using the codeml package in the PAML v4.9 framework [98]. We excluded all comparisons for which dS < 0.05 or dS > 2, as values that exceed this range are

often thought to yield unreliable estimates of rates of molecular evolution due to low statistical power and saturated substitutions, respectively. We then compared the distributions of dN/dS for each symbiont group comparison using a Wilcoxon test.

## Divergence dating

**Taxon selection.** To construct dated phylogenies for hosts and symbionts, we downloaded related genomes from NCBI. For the host divergence analysis, all of the bivalve mitochondrial genomes available as of early 2020 and four gastropod mitochondrial genomes were downloaded to serve as ingroups and outgroups, respectively (35 total taxa: 31 bivalves and four gastropods; see S4 Table). For the symbiont divergence analysis, bacterial genomes were identified for inclusion in the analysis by BLAST [75]. While residing in a relatively constrained clade of proteobacteria, these chemosynthetic symbionts do not form a monophyletic clade, have free-living relatives, are basal to more derived groups in Gammaproteobacteria, and are currently taxonomically unclassified, so it was necessary to fish out related genomes by identifying sequence homology. To do this, we aligned the nucleotide coding sequences from each one of the seven symbiont genomes we sequenced and/or analyzed against NCBI's Prokaryotic RefSeq Genomes database with blastn (-best_hit_overhang 0.1 -best_hit_score_edge 0.1 -evalue 1e-6). Based upon the diversity of hits across symbionts and genomes, we selected the top three best hits to each gene as taxa to include in the full genome divergence analysis (59 total taxa: 58 gammaproteobacteria and one alphaproteobacterium outgroup; see S5 Table).

**Multiple sequence alignment.** As bivalve mitochondrial genomes exhibit notoriously diverse structural arrangements [99], we used the whole rearrangement-aware genome aligner progressiveMauve [93] to align the molluscan mitochondrial genomes. These alignments were manually inspected and converted to fasta format in Geneious Prime (version 11.0.6+10) [100].

We identified and aligned orthologous proteins among these diverse bacterial genomes with bcgTree [101], then we back-translated the resulting amino acid alignments to nucleic acids with RevTrans [102]. As bcgTree only includes protein-coding genes, we manually extracted and aligned 16S and 23S ribosomal RNA sequences for these taxa with Mafft (using the accurate mafft-linsi setting) [103]. Although recombination clearly occurs frequently in the solemyid and mytilid symbiont populations, we decided against removing recombinant sites because doing so may exacerbate recombination-induced artifacts [104]. Finally, we concatenated these alignments with the nucleotide alignments from bgcTree/RevTrans with a custom perl script and inspected them in Geneious Prime.

**Phylogenetic inference and divergence dating.** We first inferred maximum likelihood phylogenies with RAxML (version 8.2.1, with parameters: f a -m GTRGAMMA -N 1000) [88] to verify that the taxa selected were able to resolve the relationships among hosts and among symbionts and free-living bacteria. Both mitochondrial and symbiont phylogenies were well-resolved (S2 Fig).

We inferred Bayesian phylogenies and dated node divergences for host mitochondria in Beast2 [105]. After several rounds of parameter testing to ascertain the speciation model and calibration date distribution that best fit the data (see Section 1 of S1 Text), we selected the Yule model of speciation, with a gamma distributed Hasegawa, Kishino, and Yano (HKY) model of substitution and a relaxed local molecular clock, and we calibrated dates to the base of the ingroup, Bivalvia. We used the fossil-based minimum appearance date for bivalves of 520 million years (first appearance estimated in Fossilworks [106] from fossil data in [107,108]). MCMC chains were run in duplicate until posterior probability convergence, around 8e8 steps for mitochondria. We also performed independent divergence date

estimations in RelTime [109,110] and PATHd8 [111] to compare to the Beast2 results (see Section 1 of S1 Text and S6 Table).

Given the consistency in RelTime and Beast2 estimates for mitochondria (S6 Table) and the unreasonably long run times necessary (several weeks) for symbiont dated phylogenies to reach posterior convergence in Beast2, we estimated symbiont divergence times in RelTime. We used the previously estimated divergence date for Gammaproteobacteria of 1.89 billion years (based on calibration to the cyanobacterial-caused atmospheric oxygenation event [112]) with a log-normal distribution to calibrate a relaxed local clock, using a gamma-distributed Tamura-Nei model of substitution, and allowing for invariant sites. All trees were plotted in FigTree.

## Symbiont species descriptions

Using the genomic, phylogenetic, and divergence data we generated above, we diagnosed and described the six symbiont species sequenced in this study. These classifications will be helpful in future investigations and discussions of symbiont function and diversity. Diagnoses of symbiont genera and descriptions of symbiont species are described in the Section 3 of S1 Text and listed in S3 Table.

## Parameter estimation via approximate Bayesian computation

**Simulation setup.** To simultaneously estimate the rates of horizontal transmission, effective homologous recombination rates, and the recombinant tract length, we used an approximate Bayesian computation approach. Here, we define the population-scaled per-base pair recombination rate, rho, to be equal to two times the effective population size times the per-base pair rate of gene conversion (rho = $2*N_e r$). The recombination tract length, l, is defined as the length of the gene conversion segment in base pairs. We used the bacterial sequential Markovian coalescent simulation framework, FastSimBac [113], to simulate neutral coalescence across a range of input parameters (see Section 2 of S1 Text for model proof). We drew the effective mutation rate from a log-uniform (3e-5,1e-2) distribution, the effective recombination rate from a log-uniform (1e-6,1e-2) distribution and the recombinant tract length from a uniform (1,1e5) distribution. Because the clonal frame cannot be inferred for the *Bathymodiolus* and *Solemya* symbiont populations, presumably due to the high levels of recombination relative to other bacteria, we did not supply the program with a fixed precomputed clonal frame for any simulations. In total we performed 100,000 simulations and we used subsets from each simulation to obtain summary statistics to train the variable sample size simulations.

**Summary statistics.** We selected a set of summary statistics that each incorporate some feature of the overall diversity (relevant for theta), and the overall effective recombination rate per site (rho*l). Specifically, we computed two estimators of theta, Waterson's theta [89], and pi [90], and we included the proportion of pairs of non-singleton SNPs at various genomic distances where all four possible combinations of alleles are observed in our sample. We placed the divisions between distance bins for pairwise comparisons of sites at 1e1, 1e2, 1e3, 1e4, 1e5, 1e6 base pairs.

**Model fitting via random forest regression.** We use the scikit-learn package to perform random forest regression to obtain estimates of each parameter for each endosymbiont population using the 4-gamete sites identified above and a custom Python script available on Github. First, we confirmed that our summary statistics are sufficient to accurately fit our desired population parameters using out-of-bag score during model training and for each population sample size (S9 Table). Although the score is often slightly lower for smaller sample sizes, this approach performs sufficiently well and consistently across samples for our

applications here. We additionally obtained confidence intervals for each parameter estimate using the forestsci package [114]. It should be noted that our simulations assume an equilibrium population. If this assumption is violated in a subset of the taxa that we examined, it might affect our parameter estimates. Nonetheless, it is unlikely that the large-scale differences in estimated parameters that we observe among groups, which are consistent with prior expectations, are entirely attributable to this potential bias.

**Method validation on existing datasets.**   In light of the relatively high recombination rates that we estimate, it is valuable to confirm that our approach for estimating rho and theta performs as expected. We therefore applied our method to the *Bacillus cereus* dataset [115] that has been studied in similar contexts using several related approaches [51,113,115]. In prior work with this dataset, estimates of the total impact of recombination, rho*l/theta, have varied somewhat, from approximately 3.7 [113] to 35.9 [115] and 229 [51]. Using our method, we obtain a value intermediate to these at 17.1, which indicates a moderate impact of recombination on genome evolution. This result suggests that our method is reliable and does not substantially inflate recombination rate estimates.

**Read-backed phasing to confirm recombination estimates.**   Because we used consensus symbiont genomes during model fitting, it could be possible that the high estimated rates of recombination in solemyid and bathymodiolin samples are an artefact of differential haplotype coverage across the genomes of genetically diverse symbiont chromosomes. We therefore sought to confirm our recombination rate estimates via comparing the rate of occurrence of all four possible configurations of two proximal alleles using read-backed phasing. Because each read-pair must ultimately derive from a single DNA fragment, when two alleles are observed on the same read or read pair, in the absence of errors, they must reflect an allelic combination that's observed within a single bacterial chromosome.

We therefore analyzed in aggregate all reads that overlapped two or more consensus alleles for each population and computed the fraction of all four possible sampling configurations. More specifically, we computed the proportion of reads sampled from across all reads in all individuals that contained alleles, AB, Ab, aB, and ab, for two adjacent biallelic sites with alleles A/a and B/b. To reduce the impact of sequencing errors, we recorded a site as containing all four possible configurations when all configurations were present at proportion greater than 0.05 in the total set of read pairs. We further limited comparisons to alleles where we could obtain at least 100 observations of both sites on single read pairs. We excluded the populations of vesicomyid endosymbionts from this analysis because too few polymorphic sites were present within the distances spanned by individual read pairs to confidently infer the frequencies of 4D sites. However, because samples from these populations contain virtually no within-host variation, we have little reason to doubt the accuracy of the consensus genotypes resulting from differential coverage of genetically diverse bacterial haplotypes.

Because we are sampling a much larger number of lineages than we did in analyzing the consensus genome sequences, we would expect if anything is different that we observe higher rates of sites where all four possible allele configurations are present. This is precisely what we found. Specifically, we observe higher proportions of pairs of sites where all four possible allelic combinations are represented at each distance considered and in each population in the read-backed dataset than in the consensus chromosomes (S10 Table). Furthermore, because we have placed conservative cutoffs on the proportions of read pairs required to consider a pair of sites as containing all four alleles, these values are likely underestimates of the true rates. Our observed rates of four-gamete test failures is therefore consistent with our analyses of consensus genome sequences and confirms that recombination must be common within these endosymbiont populations.

## Supporting information

**S1 Fig. Overview of the genomic data production and analysis steps used to study the population genomic processes influencing endosymbiont genome evolution.**
(TIF)

**S2 Fig. Maximum likelihood phylogenies for host mitochondria (top) and symbionts (bottom).** Groups of chemosynthetic associations are colored as in Fig 1: yellow = vesicomyids, green = solemyids, and blue = bathymodiolids. Mitochondrial and symbiont trees are rooted by gastropod and alphaproteobacterial outgroups, respectively. Scale bar = substitutions per site. Bootstrap support values indicated at nodes.
(TIF)

**S3 Fig. Within-host symbiont folded allele frequency spectra for all *B. septemdierum* and *B. childressi* intrahost samples with more than 50x and 45x Illumina sequencing coverage, respectively (see S1 Table for coverages and S8 Table for diversity statistics).**
(TIF)

**S4 Fig. Within-host symbiont folded allele frequency spectra for all *Solemya velum* and *Solemya pervernicosa* intrahost samples with more than 50x Illumina sequencing coverage (see S1 Table for coverages and S8 Table for diversity statistics).**
(TIF)

**S5 Fig. Within-host folded allele frequency spectra for all *Calyptogena fausta* and *Calyptogena magnifica* intrahost samples with at least 50x Illumina sequencing coverage (see S1 Table for coverages and S8 Table for diversity statistics).**
(TIF)

**S6 Fig. Endosymbiont inheritance modes.** Our generalized coalescent model of endosymbiont inheritance includes symbiont transmission modes ranging from strict horizontal transmission to strict vertical transmission, with mixed modes, exhibiting both horizontal and vertical strategies. The host populations (grey) undergo Wright-Fisher reproduction. Endosymbiont lineages (red and blue) either switch between host lineages or are inherited, depending on the transmission mode, until they coalesce in the same host lineage (purple).
(TIF)

**S7 Fig. The observed number of pairwise differences across a range of parameters under the endosymbiont population model described above.** Each distribution is 100 replicates with varying NH, H, and NS. The expectation following Equation 9 above is plotted as a red line and differs by less than 2 segregating sites from the observed mean for all cases investigated here.
(TIF)

**S8 Fig. Local alignments suggest that few rearrangements have occurred between the *S. velum* and *S. elarraichensis* symbiont genomes.** *S. elarraichensis* symbiont is the closest known relative of the *S. velum* symbiont, however material is exceedingly hard to obtain for this association, which occurs at a mud volcano at approximately 500–1000 m depth, and only a fragmented draft genome assembly was available. However, even these relatively short range segments reveal complete synteny (left). In comparison, over the same genomic distances, many rearrangements are evident between *S. velum* and *S. pervernicosa* (right), with the minority of segments retaining synteny.
(TIF)

**S1 Table. Sample, sequencing library, and mapping coverage information.** The second set of coverages listed for *C. fausta* apply to the libraries used for the intra-host analysis.
(XLSX)

**S2 Table. *De novo* reference assemblies were assembled with Nanopore reads and polished with Illumina data.** Illumina reads were used for individual sample genotype calling. There were no gaps (Ns) in any of the assemblies. The percent complete measure reflects how many of the 34 "essential genes" (see Materials and Methods) were found in the assembled genomes.
(XLSX)

**S3 Table. Symbiont species named in this study and named previously.** See S3 Supporting Text for full diagnoses and descriptions.
(XLSX)

**S4 Table. Taxa and accession numbers used in the mitochondrial genome phylogenetic analysis and divergence dating.**
(XLSX)

**S5 Table. Taxa and accession numbers used in the bacterial whole genome phylogenetic analysis and divergence dating.**
(XLSX)

**S6 Table. Divergence date estimates from different Beast2, TimeTree, and PATHd8 runs with different parameter values.**
(XLSX)

**S7 Table. Between-host symbiont population statistics calculated from consensus symbiont and mitochondrial genome sequences.** Random Forest (RF) theta and log10(rho*l) estimates were inferred by fitting genome-wide values of pi, Watterson's Theta, and 4-gamete sites to values generated in coalescent simulations.
(XLSX)

**S8 Table. Within-host symbiont and mitochondrial genetic diversity statistics.** Mapping coverages in S1 Table.
(XLSX)

**S9 Table. Out-of-bag (oob) scores for random-forest models for each parameter of interest, rho*l and theta, and for each sample size of endosymbiont individuals considered.** Oob scores indicate how often the trained model is able to predict known values, with perfect prediction equal to one.
(XLSX)

**S10 Table. Proportion of within-host variant sites that pass the 4-gamete test for recombination based upon read and read pair data over the given genomic intervals (constrained by Illumina library insert sizes).**
(XLSX)

**S11 Table. Comparative symbiont genome statistics and mobile element (ME) content.** MEs were identified as regions within the symbiont genomes with high sequence identity to elements in insertion sequence, phage, and integrative conjugative element databases.
(XLSX)

**S12 Table. Full list of ICEberg and ACLAME database mobile element hits with > = 90% sequence identity to endosymbiont genomic regions and genes.**
(XLSX)

**S1 Text. Supplemental text sections 1–3 for Forever young bacterial symbiont genomes.**
(PDF)

## Acknowledgments

We thank two anonymous reviewers and Emma George for their helpful comments on the manuscript and Xavier Didelot for kindly providing the *Bacillus cereus* validation dataset. For *Calyptogena* freezer samples collected previously, we thank Colleen Cavanaugh and Peter Girguis. For their assistance in collecting *Bathymodiolus* samples from the Lau Basin and Veatch Canyon, respectively, we thank the crews of the R/V *Falkor* and ROV *ROPOS* and the R/V *Atlantis* and HOV *Alvin*. We thank Peter Wilton for feedback on the symbiont coalescent model derivation.

## Author Contributions

**Conceptualization:** Shelbi L. Russell, Russell Corbett-Detig.

**Data curation:** Shelbi L. Russell, Evan Pepper-Tunick, Ashley Byrne, Jennie Ruelas Castillo, Christopher Vollmers, Roxanne A. Beinart, Russell Corbett-Detig.

**Formal analysis:** Shelbi L. Russell, Russell Corbett-Detig.

**Funding acquisition:** Roxanne A. Beinart, Russell Corbett-Detig.

**Investigation:** Shelbi L. Russell, Russell Corbett-Detig.

**Methodology:** Shelbi L. Russell, Russell Corbett-Detig.

**Project administration:** Shelbi L. Russell, Russell Corbett-Detig.

**Resources:** Russell Corbett-Detig.

**Software:** Shelbi L. Russell, Russell Corbett-Detig.

**Supervision:** Shelbi L. Russell, Russell Corbett-Detig.

**Validation:** Shelbi L. Russell, Russell Corbett-Detig.

**Visualization:** Shelbi L. Russell, Jesper Svedberg, Russell Corbett-Detig.

**Writing – original draft:** Shelbi L. Russell, Russell Corbett-Detig.

**Writing – review & editing:** Shelbi L. Russell, Jesper Svedberg, Christopher Vollmers, Roxanne A. Beinart, Russell Corbett-Detig.

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
