## [Decision Letter · Decision Letter 0]

6 Jan 2020

Dear Dr Russell,

Thank you very much for submitting your Research Article entitled 'Horizontal transmission and recombination maintain forever young bacterial symbiont genomes' to PLOS Genetics. Your manuscript was fully evaluated at the editorial level and by three independent peer reviewers with complementary expertise. The reviewers appreciated the attention to an important problem, but raised some substantial and thoughtful concerns about the current manuscript. Based on the reviews, we will not be able to accept this version of the manuscript, but we would be willing to review again a much-revised version. We cannot, of course, promise publication at that time.

Should you decide to revise the manuscript for further consideration here, your revisions should address all the specific points made by each of the three reviewers. We will also require a detailed list of your responses to the review comments and a description of the changes you have made in the manuscript.

If you decide to revise the manuscript for further consideration at PLOS Genetics, please aim to resubmit within the next 90 days, unless it will take extra time to address the concerns of the reviewers, in which case we would appreciate an expected resubmission date by email to plosgenetics@plos.org.

[LINK]

We are sorry that we cannot be more positive about your manuscript at this stage. Please do not hesitate to contact us if you have any concerns or questions.

Yours sincerely,

Xavier Didelot

Associate Editor

PLOS Genetics

Kirsten Bomblies

Section Editor: Evolution

PLOS Genetics

Reviewer's Responses to Questions

**Comments to the Authors:**

Reviewer #1: This manuscript by Russell et al is a very interesting investigation of the evolution of bacterial endosymbionts of molluscs. The analyses are varied and complex, providing an overview of three different types of endosymbiotic association, and how they affect the evolution of the bacteria involved. The work has the potential to be an important contribution to the literature on fundamental processes in bacterial evolution, but the methodological novelty of the study means there is substantial clarification and validation required before the manuscript is ready for publication.

The main points that need to be addressed are:

(1) The title makes reference to these bacteria maintaining “forever young” genomes. However, there is not much evidence for the long-term stability of the observed genotypes. S5 Table suggests there is not much diversity within each bacterial species (with very little diversity in the Calyptogena populations), and S7 Fig suggests much of this is likely to be due to be the result of a small number of long branches. Using the most relevant molecular clock data available, the authors should estimate (A) the time to the most recent common ancestor of the sequenced bacteria (eliminating the diversification through recombination as the coalscent analysis should allow, but it should not make much difference for Calyptogena), and (B) compare this to the estimated times at which the host species diverged. This should estimate the duration over which the symbiosis can be analysed. An alternative hypothesis to that proposed by the authors is that the ability for horizontal acquisition of these bacteria by hosts makes it possible for environmental bacteria to invade the niche and displace the resident symbionts (the “symbiont replacement” mentioned in the Introduction), explaining the observed data (including any most recent common ancestors that substantially post-date the divergence of hosts) without the genomes being “forever young”. Rather, this would support the model in which the genomes of the symbionts degrade, and are intermittently replaced by new species from the environment.

(2) The authors state “the other [non-vesicomyid] symbiont genomes are highly structurally dynamic”. However, Figure 4E demonstrates that these non-vesicomyid symbionts are polyphyletic – it is not surprising that there are many changes distinguishing the aligned pairs. The authors need to provide evidence that these representatives have had an endosymbiont lifestyle over the full period of their divergence (are the related species all endosymbionts?), or make clear they have become endosymbionts in parallel, and the genetic divergence has occurred outside of these hosts. The rest of the paragraph is confusing, not least owing to a lack of specificity (“It is”, “these groups”). The authors state that “This could proceed by iteratively locking deletions (initially caused by recombination… With recombination events occurring so rarely in these endosymbiont genomes, natural selection is unable to efficiently purge deleterious deletions. Inversions, which can be highly mutagenic by inverting the translated strand, inducing replication-transcription machinery collisions [33], may be nearly absent due to their high fitness costs.” – does recombination increase or decrease the rate at which deletions accumulate, and is selection weak (enabling deletions) or strong (preventing inversions)? Additionally, when stating “This is consistent with a combination of free-living and host associated periods”, they should specify whether this refers to the species current lifestyles (which would conflict with the description of these bacteria as obligate symbionts in the abstract, though this phrase is not repeated in the main paper), or over the history of the observed species.

(3) The authors refer to recombination repeatedly, but never specify whether they mean homologous recombination (cross over or gene conversion), horizontal gene transfer, or mobile elements. They should make clear which they mean in each analysis. In the S2 Supporting Text, they refer to “When a recombination event occurs on a genealogy, it creates an additional lineage to the right of the recombination breakpoint.” – this sounds like a crossover event, which is not a typical form of recombination in bacteria. The text does not specify the interpretation of rho, rho*l or r/m sufficiently. Most bacterial geneticists would interpret r/m as being the ratio of single nucleotide variants occurring through recombination to those occurring through point mutation (e.g. Feil et al 1999 Mol. Biol. Evol. 16(11):1496–1502), yet the r/m for C. fausta is 3625 (it is rarely over 10 for any bacterial species except the most rapidly recombining), despite it having a low rate of recombination and only 815 segregating sites. It seems likely the analysis is prone to false positive inferences of recombination – the calculated rates are high, and even bacteria lacking recA were found to undergo recombination. The authors should report the estimated rate of recombination for an analogous analysis of the mitochondrial DNA sequences to demonstrate the method is robust to false positives – perhaps additionally the method could be validated against datasets where r/m has been calculated by alternative methods?

Minor criticisms:

• S1 Table should be a spreadsheet with accession numbers for the included samples

• Figure 3B needs labelled scale bars – the trees should be presented more clearly, with leaf tip names clearly readable. Data from S7 Fig should be included as well.

• Figure 4E should be a separate, enlarged figure, with leaf tip names clearly readable

• The supplementary tables need more detailed legends, particularly S4 Table, S6-S8 Tables

• The authors need to explain how the p values on Pg. 9 Ln. 164-167 were calculated

• Why were genes with extreme dN/dS ratios excluded, rather than genes with small numbers of mutations? “Wilcox test” should be “Wilcoxon test”.

• The four gamete test should be more thoroughly explained.

• “Nanopore” and “Illumina” are sometimes not capitalized. There are contractions in the main and supplementary text (e.g. “that’s”).

Reviewer #2: Summary:

This study investigates the genome evolution of gammaproteobacterial symbionts from three groups of bivalves and provides an interesting symbiont study system, with the three host groups exhibiting different modes of symbiont transmission: horizontal, vertical and mixed mode transmission. Surprisingly, one symbiont group with reduced genomes has maintained a relatively stable genome size that is likely due to a moderate genetic recombination rate compared to others like insect symbionts with severely reduced genomes. This is a great study, but there are several aspects that need clarification and the authors have focused on the idea that the symbionts can maintain stable genome size indefinitely without much evidence or a discussion of other possible outcomes.

Major Comments:

Although using “forever young” in the title is catchy, this may not be true as stated in lines 186 – 187. At this moment in time, vesicomyid symbionts may be able to slow genome degradation and maintain an intermediate genome size, but the processes of Muller’s ratchet are still at work and continued genome reduction/degradation is a likely outcome. In which case, lines 206 – 207 also need to be rephrased, especially the term “indefinitely”.

I found it hard to follow the population genetics discussion without having more ecological context of the bivalve systems. Please include a short description of the transmission modes (how many symbionts are transferred during vertical transmission, how host switching occurs, how the hosts gain symbionts from the environment, are the symbionts obligate intracellular bacteria or facultative, etc.).

An “organelle-like state” is mentioned throughout the paper, but this term is highly debated. What is your definition of organelle-like?

Line 685 (Figure 1): It’s interesting that the vesicomyid symbiont and mitochondrial phylogenies are discordant since those symbionts are vertically transmitted. Can you explain more about why there are “sufficient amounts of horizontal transmission” occurring in vesicomyid symbionts? Also, please explain the difference between host-switching and horizontal transmission in vesicomyid symbionts. Can these symbionts be found in the environment (e.g. free-living) or do they become transferred directly from other non-related hosts?

Another aspect that I found interesting is that the vesicomyid symbionts have the smallest genomes of the bivalve symbionts but evolved from their free-living ancestors much later (66 mya) than the solemyid symbionts (400 mya). Therefore, vesicomyid symbiont genome reduction occurred relatively quickly compared to the solemyid symbionts. Also, the solemyid symbionts seem to have evolved from free living ancestors multiple times whereas the vesicomyid symbionts arose only once. Are these differences due strictly to transmission modes or are there other factors involved?

Also according to the symbiont phylogenetic tree, the solemyid symbionts evolved multiple times from free living ancestors and the mytilid symbionts evolved at least twice. Have these hosts replaced symbionts multiple times with free-living bacteria that eventually became new symbionts and thus, a range of symbiont genome sizes exist (at least in the solemyid symbionts)? This has happened in other symbiosis systems like mealybugs (Husnik et al. 2016), lice (Smith et al. 2013) and the protist, Euplotes (Boscaro et al. 2017) so it seems possible that these host groups also replaced the symbionts when deleterious mutations were fixed in symbiont populations, especially mutations in carbon fixation pathways.

Although a few examples of other symbiosis systems are included, a quick summary about the processes that drive genome reduction in other symbionts (e.g. insect symbionts) compared to those in the bivalve symbionts is needed. What are some differences and/or similarities between the systems (selection, drift, transmission, age of the symbiosis, etc.)? This would be a nice place to discuss why the vesicomyid symbionts have maintained a stable genome size whereas other systems have severely reduced symbionts. What are the processes involved and why would the vesicomyid symbionts have a different outcome?

Minor comments:

Please introduce the study system in the abstract since it is not mentioned until the last introduction paragraph.

(Line 33) Eukaryotes have not existed for the “history of life on Earth”

(Line 75 – 76) There are known marine symbiont genomes less than 1 Mb: bacterium AB1 in the marine invertebrate, Bugula neritina (Miller et al. 2016); Spiroplasma holothuricola in sea cucumbers (He et al. 2018); alphaproteobacteria in the marine protists, diplonemids (George et al. 2019)

(Figure 2) Switch Panel C and D labels so that the labels go in alphabetical order.

Reviewer #3: The authors ask an important and timely question about how the long-term stability of host-symbiont associations influences the symbiont genome evolution. They address the question by comparing some genomic properties of chemosynthetic gill-inhabiting bacterial symbionts in seven species representing three families of bivalves, with different modes of the symbiont transmission.

The authors have generated substantial amounts of novel quality data. I feel that the presented work has the potential of adding substantially to our understanding of symbioses of these bivalves, and symbioses more generally. However, I have several major concerns.

The presented data focus on seven bivalve species from three families with different symbiont transmission modes. Considering this, I feel that the author's claim of "testing the canonical theory" "by sampling marine endosymbionts that range from primarily vertical to strictly horizontal transmission", and that "These results validate long-standing but untested theory and demand a reinterpretation of the vast diversity of symbioses..." is an overstatement. I argue that major evolutionary theories relevant to a wide range of systems cannot be adequately tested using only three independent data points (=families). While the theoretical background provided in the Introduction is solid and relevant, I strongly recommend that the authors shift the focus of this study to what the study actually is: a comparative description of symbioses in three families of bivalves.

In an apparent attempt to make the paper appear relevant to a broad audience, the authors hid the information of what they actually studied. I found it striking that bivalves - the animal group that was studied - is not mentioned in the title, abstract, or author summary, not even once! Furthermore, the authors provided very little introductory background information for these bivalves or their symbionts [lines 100-102]. As the only reference to support the statement that the three bivalve families differ in their modes of symbiont transmission, they provided a meta-analysis of >500 host-microbe symbioses, and some of the results of the current study. More information about the nature and old age of these symbioses is provided at the end of the Discussion - not a place where readers typically look early on. Without that background information or experience with bivalve symbioses, I have found it difficult to understand what the authors have done and why, and what the results mean. Some of the key questions that the Introduction should address are:

* What is known about bivalve symbioses? What are their functions and localizations? How specific are they?

* Are the focal families obligately associated with symbionts?

* In their gills, do bivalves host single symbiont strains, or more complex multi-strain or multi-species microbiota?

* How these symbionts transmit - what did we know before the current study?

* What has been known about the functions of these symbionts?

In my opinion, the authors need to develop the second half of the Introduction around these questions, explaining how their presented work fills gaps in our existing knowledge.

Another major weakness of the study is the way information is organized in the article. I struggled to understand what exactly the authors did, how they did it, what was the reasoning, and which of the findings are new. I provided one example in the previous paragraph: the authors state that they have compared bivalve families with different symbiont transmission modes, citing a meta-analysis and some parts of the current study. Then, how much has been known about the transmission in the selected groups of bivalves? Differences in transmission modes ... is this an entirely new finding, or has there been prior data? After completing this review, I am still not sure.

I realize that the journal requires that the methodological details are only provided in the last section or/and in the supplement. Despite this, I would expect to find a summary of what the authors did at the end of the Introduction, or perhaps at the beginnings of paragraphs corresponding to Results. That was not the case here. It took me a substantial amount of effort (jumping between sections of the main text and the supplement) to understand that the authors have sequenced and analyzed gill metagenomes from multiple individuals from a total of eight populations of seven bivalve species. I further understood that for one individual per species, a reference symbiont + mitochondrial genome was assembled, facilitating the use of lower-coverage data for the symbiont diversity comparisons within and across hosts. Despite my background in invertebrate symbiont diversity and genomics, I failed to understand some of the analyses. A summary of the experimental setup would have greatly helped me follow the text. Please make it easy for readers to understand what you did!

I am puzzled by some of the findings of the study, and their interpretation. The authors claim in the Abstract that they compare symbiotic systems "that range from primarily vertical to strictly horizontal transmission". However, the host-symbiont phylogeny comparison (Fig. 1C) does not make it clear that the differences among host species are significant. At the same time, I am surprised that when sampling replicate individuals from a population, the authors discovered substantial mitogenomic diversity. That is not a pattern I have seen in insects; is it expected in bivalves? The authors seem to be using the smaller genome sizes of vesicomyid symbionts, lower recombination rates and other genomic features as a secondary confirmation of the deducted mode of transmission. Once again, was the mode of transmission known before the study? If not, then what is the likelihood that the observed patterns are due to biological differences among hosts, for example in the intra-population diversity?

I have not been familiar with folded allele frequency spectrum analyses, and I struggled to comprehend the plots in Figs 2-3 and S2-S4. Perhaps the authors could explain and discuss the analyses and patterns in a way that makes it easier for biologists not familiar with that particular method to understand the expectations and interpret patterns from these plots?

In the main text, the authors talk about vesicomyid, mytilid/bathymodiolin, and solemyid symbionts. In the figures, they provide Latin names of host species. I found it somewhat difficult to match and navigate the sets of names, and ended up searching multiple times which species belongs to which group. Perhaps the authors could think of ways of editing figures in a way that would help readers make the connection?

Finally, the authors use the genomic data for phylogenetic reconstructions. The addition of functional / contents analysis would have made the current story much stronger.

To sum up, I feel that the dataset holds strong promise for a publication. I strongly recommend that the authors present these data in the context of filling gaps in the understanding of bivalve symbioses. The systematic testing of whether the symbiont genomic patterns align with expectations for a given transmission mode would be a useful addition, but should not be the main focus. The authors need to explain in a clear and accessible way what was the state of the field, and which of the findings are new. They also need to state clearly what they did and why, how the findings complement existing knowledge, and what the results mean.

Specific comments:

Lines 210-211: The authors wrote: "These results validate long-standing but untested theory and demand a reinterpretation of the vast diversity of symbioses...". I am finding the claim that "the vast diversity of symbioses" should be reinterpreted based on data for three families that largely conform to the expectations quite arrogant.

Lines 231-233: Name kit/enzyme manufacturers

Lines 233-234: "to uniquely label both i5 and i7 indexes for each sample that was sequenced on a single lane of a Hiseq4000. We sequenced a total of four lanes of Hiseq4000"... Before I read into the supplement, I was quite confused. I initially understood that six spp. were sequenced, each filling its own lane. Change the wording.

Lines 297-298: The authors wrote "After producing endosymbiont genome assemblies and host mitochondrial genome assemblies for each host/endosymbiont population..." . This is misleading and confusing: the assemblies were provided for a single host individual per population.

Fig. 1C. What genes are the trees based on? Please provide scale bars. What are the outgroups?

Fig. 4E. The sample labels are provided, but the font size makes them virtually illegible.

Table S1. Why are the genome coverage values for nanopore reads provided as "na"?

Table S2. What is the advantage of providing the genome, scaffold etc. sizes in the format "2.37E+06" as opposed to the actual values?

In Table S3, the authors provide the list of "Symbiont species named in this study and named previously", including newly proposed names. The table is only referenced once in the main manuscript, but no information about that nomenclatural aspect is provided.

Also, I am not a Latin expert, but I believe that the proposed symbiont names are incorrect grammatically. The generic names should be nouns in the nominative, singular form, with endings corresponding to declension - but as far as I can tell, several of the generic names proposed here do not conform to these conventions. Please consult the bacterial nomenclature rules.

**Have all data underlying the figures and results presented in the manuscript been provided?**

Reviewer #1: No: It is not yet possible to tell if the data are all present in the databases, but a project accession number is provided by the authors, who indicate the data will be released on publication.

Reviewer #2: Yes

Reviewer #3: No: The authors state that the genomic data will be published at NCBI upon article acceptance

PLOS authors have the option to publish the peer review history of their article (what does this mean?). If published, this will include your full peer review and any attached files.

Reviewer #1: No

Reviewer #2: Yes: Emma E. George

Reviewer #3: No

---

## [Decision Letter · Decision Letter 1]

15 May 2020

Dear Dr Russell,

Thank you very much for submitting your Research Article entitled 'Horizontal transmission and recombination maintain forever young bacterial symbiont genomes' to PLOS Genetics. Your manuscript was fully evaluated at the editorial level and by independent peer reviewers. The reviewers appreciated the attention to an important topic but identified some aspects of the manuscript that should be improved.

We therefore ask you to modify the manuscript according to the review recommendations before we can consider your manuscript for acceptance. Your revisions should address the specific points made by each reviewer.

[LINK]

Yours sincerely,

Xavier Didelot

Associate Editor

PLOS Genetics

Kirsten Bomblies

Section Editor: Evolution

PLOS Genetics

Reviewer's Responses to Questions

**Comments to the Authors:**

Reviewer #1: I thank Russell et al for their thoughtful and extensive response to my previous comments. The phylodynamic analyses in Figure 5 are excellent, and a valuable addition – to me, it would seem helpful to have this as Figure 1, as it allows a non-specialist reader to immediately understand how divergent the host and endosymbionts are, and the durations of their associations. The validation of the recombination analysis method is also nicely done, although there are a couple of aspects it would be useful to address in point 2 below.

The remaining problem is not concerning the data, but rather the interpretation of the link between recombination, selection and genome structure. Given the authors’ answers to my first review, there are three questions that it seems important to address:

(1) As the authors state all recombination they refer to is homologous recombination, is the rate high relative to other species? The authors state, “these populations have some of the largest effective recombination rates ever reported for bacteria” – this is unaltered from their first draft. Yet the authors state in their response that they “conflated” rho*l/theta with r/m (with the former giving higher estimates of the impact of recombination) – yet the cited Vos & Didelot study reports r/m, making a comparison of these different statistics very difficult. Given their novel method predicts recombination makes a bigger impact than mutation in the Calyptogena magnifica symbiont, despite it lacking the requisite homologous recombination machinery, can the authors validate this claim, and provide a robust comparison with other bacterial species, using existing software (e.g. ClonalFrameML, chromosome painting or Gubbins)? I am also confused about the statement in the response, “When a gene conversion event occurs on a genealogy, it creates an additional lineage within the converted segment to the right and to the left of the recombination breakpoints.” Why is an additional lineage created either side of a gene conversion event – both the upstream and downstream regions are part of the same clonal frame?

(2) Do the authors think recombination prevents, or causes, genome degradation? The Results section ends, “Thus, ample time has passed for these symbiont genomes to erode, but horizontal transmission and recombination have likely prevented it.” – an inference emphasised in the Conclusion – which suggests to the reader that recombination directly limits genome degradation. Yet in the Results section, the authors suggest a “recombinational processes may partially underlie genome erosion”, as “a rare recombination event induces a deletion that drifts to high frequency” – which is the process of genome degradation. As the authors say in their response, “Whenever we refer to recombination, we mean homologous recombination”, this suggests faster homologous recombination (as seen in some of these endosymbionts) should drive faster genome degradation. Can deletions not be spontaneous, rather than occurring through recombination? Might selection not be more important than recombination in the observed differences?

(3) To what extent is the study describing the evolution of symbionts? The addition of the tree is excellent, and highlights those comparisons between endosymbionts that are sister taxa (and can reasonably be assumed to have maintained an endosymbiotic lifestyle over the period of their divergence), and those that are distantly related, and therefore are likely to have evolved as free-living bacteria over some proportion of their divergence. The authors accurately refer to “Highly divergent solemyid symbionts” in the Results, but it would be more helpful to the reader to point out that it cannot be assumed the divergence occurred as symbionts (unless the many related taxa have all independently evolved to be free living), particularly given the pairwise comparison in Figure 4C is contrasted with that in 4D, where the compared isolates are a single clade, the most recent common ancestor of which was likely an endosymbiont.

I reiterate that I think this is an interesting and valuable study, but given its complexity, it is important to clarify and assess the central message.

Minor points:

In some places in the updated text, “mya” is just “my”.

Reviewer #2: The authors improved the clarity of the manuscript and added a nice analysis of the estimated divergence dates for both the hosts and symbionts. They also answered and implemented all of my questions/suggestions which I appreciated. However, I still believe the ‘forever young’ term in the title is stretching it a bit since this term only refers to the symbionts up to this point in time. It’s difficult to predict what will happen to the symbionts in the future, especially if recombination rates and horizontal transmission decrease. Therefore, they may not be ‘forever young’. I still think this paper is invaluable to the symbiosis field, and should be accepted since very few studies on symbiont population genetics exist from hosts other than insects.

Reviewer #4: This study by Russell et al. uses a population and comparative genomics approach to examine differences in the endosymbiont population dynamics among 3 families of marine bivalves with different modes of symbiont transmission. Each symbiosis is independently evolved with endosymbionts providing chemosynthetic carbon fixation. Overall I like the study and I think it represents a significant contribution to the field of marine bacterial endosymbioses. The data generated represents a valuable resource to the broader scientific community. Additionally, after reading previous reviews I feel like the authors have addressed the most significant issues in the manuscript. I have comments.

It isn’t clear to me (and this could be because I am less familiar with marine endosymbionts) why there seems to be an expectation that these symbiont genomes should undergo the same degree of erosion as insect endosymbionts? 1) The host is gutless, and relies entirely on the symbiont for ALL of its nutritional needs, not just a handful of essential amino acids. This necessarily implies the lower bound for genome size will be much higher than for an obligate terrestrial endosymbiont, where in drift will rule the drift/selection balance for a much higher fraction of the genome. 2) Line numbers. 3) Drift is much stronger in terrestrial endosymbionts that are never exposed to an outside environment. It seems marine symbionts, even in the case of mostly vertical transmission, are still more exposed to the environment as they are essentially processing the immediate environment for the host. Additionally, given rates of horizontal transfer, they clearly have to survive in some form outside the host. Whether, the transfer events are mediated through, burst, unfertilized eggs, the contents of which are then filtered by the gills of a neighbor or otherwise, this is far beyond the level of exposure transovarially, internally fertilized terrestrial arthropod symbionts will experience. I feel the more apt comparison is among ectosymbionts of terrestrial arthropods. 4) Line numbers. 5) Despite these points, as the previous reviewer pointed out, Mueller’s ratchet will ratchet more in the vertically transmitted symbionts, and the data support this (dN/dS, minor allele frequency spectrum). However, the maintenance of diverse coding content likely wouldn’t persist for 10s or 100s of millions of years if it weren’t needed, thus I see degradation to an organellar-like state as somewhat less likely than what occurs in other systems. Perhaps the most surprising aspect is that host-symbiont cospeciation and associations persist for as long as they do without replacements. 6) Line numbers. 7) It could likely be argued that the host takes over much of the workload as the symbiont degenerates, as seen in other systems. Delegation of these functions presumably stops at carbon fixation +other complex nutritional provisioning steps (like amino acid synthesis). The point of the prior reviewer about the remaining coding capacity is relevant in this context.

Other comments.

Figure 1C- Fst as well as other data being what it is among the symbionts associated with these bivalves, I have no doubt there is gene flow between hosts as this figure is meant to convey. However, with rampant polytomies in the trees, the degree of transfer might not be nearly as prevalent as this figure suggests (I see several ways of making it more concordant at a glance). I understand that you point this out in the text, and that this figure is more or less supposed to be a toy model to make a point about concordance, however, this figure doesn’t support the point that gene flow erodes host/symbiont concordance because it tells us little about the relationships that are supposedly “discordant” due to gene flow. Also, gene flow and recombination erode our ability to produce fully resolved phylogenies. These are trees made from populations, with a single consensus likely representing the major allele. With lots of gene flow and recombination it isn’t likely one would resolve these in any meaningful way at the population level but a polymorphism aware model (such as the one implemented in IQ-Tree2) run on allele frequency data would be a big improvement. The model allows for fixation of heterozygosity with Watterson’s theta or specifying the distribution from which to draw states. As for the mitogenome trees, I would use the ML tree without collapsing branches (maybe partition by codon position in the future), low support is expected with so few segregating sites and a single locus, but knowing the “best” hypothesis is still more informative than polytomies.

Figure 2C- The solemyids seem odd. I understand by looking at the supplementary that this is meant to represent the fact that they are intermediate. However, some individuals of S. pervernicosa almost appear to be in balancing selection? This appears to hold for the minor allele freq spectrum of individuals as well as Tajima’s D. I guess this would be the case of drift or an individual that had gene flow restricted for a number of generations and then recently had a transfer event. I guess my question is whether you see in nonsense mutations on one haplotype background that is compensated for in the alternative haplotype and vice versa? Since you have nanopore and read backed phasing data, this might be interesting to look at.

I have attached a version with line numbers that has additional comments..

**Have all data underlying the figures and results presented in the manuscript been provided?**

Reviewer #1: No: The accession codes are not included in the manuscript - the authors state they will be submitted only after acceptance. Given the policies of PLOS journals, it would seem appropriate to request evidence of submission at this stage.

Reviewer #2: No: The authors state that the genomes will be submitted to NCBI directly before publication.

Reviewer #4: Yes

PLOS authors have the option to publish the peer review history of their article (what does this mean?). If published, this will include your full peer review and any attached files.

Reviewer #1: No

Reviewer #2: No

Reviewer #4: No

---

## [Editor Report · Decision Letter 2]

16 Jun 2020

Dear Dr Russell,

We are pleased to inform you that your manuscript entitled "Horizontal transmission and recombination maintain forever young bacterial symbiont genomes" has been editorially accepted for publication in PLOS Genetics. Congratulations!

Yours sincerely,

Xavier Didelot

Associate Editor

PLOS Genetics

Kirsten Bomblies

Section Editor: Evolution

PLOS Genetics

Comments from the reviewers (if applicable):

**Data Deposition**

http://datadryad.org/submit?journalID=pgenetics&manu=PGENETICS-D-19-01871R2

**Press Queries**

---

## [Editor Report · Acceptance letter]

21 Jul 2020

PGENETICS-D-19-01871R2 

Horizontal transmission and recombination maintain forever young bacterial symbiont genomes 

Dear Dr Russell, 

We are pleased to inform you that your manuscript entitled "Horizontal transmission and recombination maintain forever young bacterial symbiont genomes" has been formally accepted for publication in PLOS Genetics! Your manuscript is now with our production department and you will be notified of the publication date in due course.

With kind regards,

Kaitlin Butler

PLOS Genetics

On behalf of:
